# Don't Generate Me:
# Training Differentially Private Generative Models with Sinkhorn Divergence

**Tianshi Cao**[1,2,4]    **Alex Bie**[3*]    **Arash Vahdat**[4]    **Sanja Fidler**[1,2,4]    **Karsten Kreis**[4]

[1]University of Toronto    [2]Vector Institute    [3]University of Waterloo    [4]NVIDIA

tianshic@nvidia.com,    yabie@uwaterloo.ca,    {avahdat,sfidler,kkreis}@nvidia.com

## Abstract

Although machine learning models trained on massive data have led to break-throughs in several areas, their deployment in privacy-sensitive domains remains limited due to restricted access to data. Generative models trained with privacy constraints on private data can sidestep this challenge, providing indirect access to private data instead. We propose DP-Sinkhorn, a novel optimal transport-based generative method for learning data distributions from private data with differential privacy. DP-Sinkhorn minimizes the Sinkhorn divergence, a computationally efficient approximation to the exact optimal transport distance, between the model and data in a differentially private manner and uses a novel technique for controlling the bias-variance trade-off of gradient estimates. Unlike existing approaches for training differentially private generative models, which are mostly based on generative adversarial networks, we do not rely on adversarial objectives, which are notoriously difficult to optimize, especially in the presence of noise imposed by privacy constraints. Hence, DP-Sinkhorn is easy to train and deploy. Experimentally, we improve upon the state-of-the-art on multiple image modeling benchmarks and show differentially private synthesis of informative RGB images. Project page: https://nv-tlabs.github.io/DP-Sinkhorn.

## 1    Introduction

Modern machine learning (ML) algorithms and their practical applications (e.g. recommender systems [1], personalized medicine [2], face recognition [3], speech synthesis [4], etc.) have become increasingly data hungry and the use of personal data is often a necessity. Consequently, the importance of privacy protection has become apparent to both the public and academia.

Differential privacy (DP) is a rigorous definition of privacy that quantifies the amount of information leaked by a user, participating in a data release [5, 6]. The degree of privacy protection is represented by the privacy budget. DP was originally designed for answering queries to statistical databases. In a typical setting, a data analyst (party wanting to use data; e.g. a healthcare company) sends a query to a data curator (party in charge of safekeeping the database; e.g. a hospital), who makes the query on the database and replies with a semi-random answer that preserves privacy. Responding to each new query incurs a privacy cost. If the analyst has multiple queries, the curator must subdivide the privacy budget to spend on each query. Once the budget is depleted, the curator can no longer respond to queries, preventing the analyst from performing new, unanticipated tasks with the database.

Generative models can be applied as a general and flexible data-sharing medium [7, 8], sidestepping the above problems. In this scenario, the curator first encodes private data into a generative model;

---

*Work done during internship at NVIDIA.

then, the model is shared with the analyst, who can use it to synthesize similar yet different data from the training data. This data can be used in any way desired, such as for data analysis or to train specific ML models. Unanticipated novel tasks can be accommodated without repeatedly interacting with the curator, since the analyst can easily generate additional synthetic data as required.

Furthermore, it has been observed that generative models can reveal critical information about their training data [9, 10]. For example, Webster et al. [9] found that modern GANs trained on images of faces produce examples that greatly resemble their training data, thereby leaking private information. Hence, the generative model must be learnt with privacy constraints to protect the privacy of individuals contributing to the database.

Differentially private learning of generative models has been studied mostly using generative adversarial networks (GANs) [7, 11, 12, 13, 14]. While GANs in the non-private setting can synthesize complex data such as high definition images [15, 16], their application in the private setting is challenging. This is in part because GANs suffer from training instabilities [17, 18], which can be exacerbated by adding noise to the GAN's gradients during training, a common technique to implement DP. Hence, GANs typically require careful hyperparameter tuning. This goes against the principle of privacy, where repeated access to data need to be avoided [19].

In this paper, we propose *DP-Sinkhorn*, a novel method to train differentially private generative models using a semi-debiased Sinkhorn loss. DP-Sinkhorn is based on the framework of optimal transport (OT), where the problem of learning a generative model is framed as minimizing the optimal transport distance, a type of Wasserstein distance, between the generator-induced distribution and the real data distribution [20, 21]. DP-Sinkhorn approximates the exact OT distance in the primal space using the Sinkhorn iteration method [22]. Furthermore, we propose a novel semi-debiased Sinkhorn loss to optimally control the bias-variance trade-off when estimating gradients of this OT distance in the privacy preserving setting. Since our approach does not rely on adversarial components, it avoids any training instabilities and removes the need for early stopping (stopping before catastrophic divergence of GANs, as done, for example, in [15]). This makes our method easy to train and deploy in practice. To the best of our knowledge, DP-Sinkhorn is the first fully OT-based approach for differentially private generative modeling.

In summary, we make the following contributions: (i) We propose DP-Sinkhorn, a flexible and robust optimal transport-based framework for training differentially private generative models. (ii) We demonstrate a novel technique to finely control the bias-variance trade-off of gradient estimates when using the Sinkhorn loss. (iii) Benefiting from these technical innovations, we achieve state-of-the-art performance on widely used image modeling benchmarks for varying privacy budgets, both in terms of image quality (as measured by FID) and downstream image classification accuracy. Finally, we present informative RGB images generated under strict differential privacy without the use of public data, with image quality surpassing that of concurrent works.

## 2   Related Works

The task of learning generative models on private data has been tackled by many prior works. The general approach is to introduce privacy-calibrated noise into the model parameter gradients during training. A long line of works have explored combinations of GANs and differential privacy. DP-GAN [7] first introduced the idea of combining differential privacy with GANs in a simple scheme where DPSGD [23] is used when updating the generator. This is followed up a year later by dp-GAN [11], which adds a decaying clipping threshold that heuristically matches the decreasing gradient magnitude during training. DP-CGAN [24] adds class conditioning to DPGAN for generation of conditional data. PATE-GAN [25] adopts the PATE framework for generative learning by using PATE [26] to train a private student discriminator; only generated images are scored by this student discriminator to train the generator. This work is improved by G-PATE [27], which uses random projections and gradient quantization to directly aggregate discriminator gradients for updating the generator. Importantly, G-PATE makes the point that only the generator is released in the DP generative learning task, thus a large (∼1000s) ensemble of non-private discriminators can be used to train a private generator. GS-WGAN [13] brings this idea back to DPSGD-based training, in which the gradient aggregate from an ensemble of discriminators is processed by the Gaussian mechanism. Unlike DP-GAN, this is performed on the image gradient, which has fewer dimensions than the parameter gradient. Datalens [14] further improves upon G-PATE by introducing TopAgg—a three

step gradient compression and aggregation algorithm, which provides stable, quantized discriminator gradients at a low privacy cost.

It is well documented that GANs are unstable during training [17, 18] due to the non-optimality of the discriminator producing large biases in the generator gradient [20]. This problem is critical in the context of DP, where the imposed gradient noise can increase training instabilities and interaction with private data (for example during hyperparameter tuning) should be limited. Our approach circumvents these issues by not relying on adversarial learning schemes. Furthermore, state-of-the-art methods [13, 14] rely on training a large number of discriminators to take advantage of the subsampling property of differential privacy. This hinders their practical usefulness as the discriminators require large amounts of GPU/TPU memory. In contrast, only a single generator network is trained in DP-Sinkhorn, making our approach more amenable to various hardware configurations.

Other generative models have also been studied in the DP setting. [28] partitions the private data in clusters and learns separate likelihood-based models for each cluster. [29] uses Maximum Mean Discrepancy with random Fourier features. While these works do not face the same stability issues as GANs, their restricted modeling capacity results in these methods mostly learning prototypes for each class. DP-Sinkhorn is better at using the modeling capacity of neural networks to produce high utility synthetic data while preserving privacy. Lastly, while [30] produced strong empirical results, their privacy analysis relies on the use of Wishart noise on sample covariance matrices, which has been proven to leak privacy [31]. Hence, their privacy protection is invalid in its current form.

## 3 Background

### 3.1 Notations and Setting

Let $\mathcal{X}$ denote a sample space, $\mathcal{P}(\mathcal{X})$ all possible measures on $\mathcal{X}$, and $\mathcal{Z} \subseteq \mathbb{R}^d$ the latent space. We are interested in training a generative model $g : \mathcal{Z} \mapsto \mathcal{X}$ such that its induced distribution $\mu = g \circ \xi$ with noise source $\xi \in \mathcal{P}(\mathcal{Z})$ is similar to observed $\nu$ through an independently sampled finite sized set of observations $D = \{\mathbf{y}\}^N$. In our case, $g$ is a trainable parametric function with parameters $\theta$.

### 3.2 Generative Learning with Optimal Transport

Optimal Transport-based generative learning considers minimizing variants of the Wasserstein distance between real and generated distributions [20, 21]. Two key advantages of the Wasserstein distance over standard GANs, which optimize the Jensen-Shannon divergence [32], are its definiteness on distributions with non-overlapping supports, and its weak metrization of probability spaces [33]. This prevents collapse during training caused by discriminators that are overfit to training data.

The OT framework can be formulated in either the primal or dual formulation. In WGAN and variants [33, 34, 35], the dual potential is approximated by an adversarially trained discriminator. These methods still encounter instabilities during training, since the non-optimality of the discriminator can produce arbitrarily large biases in the generator gradient [20]. The primal formulation involves solving for the optimal transport plan—a joint distribution over the real and generated sample spaces. The distance between the two distributions is then measured as the expectation of a point-wise cost function between pairs of samples as distributed according to the transport plan.

In general, finding the optimal transport plan is a difficult optimization problem. The entropy-regularized Wasserstein distance (ERWD) imposes a strongly convex regularization term on the Wasserstein distance, making the OT problem between finite samples solvable in linear time [36]. Given a positive cost function $c : \mathcal{X} \times \mathcal{X} \mapsto \mathbb{R}^+$ and $\lambda \geq 0$, the ERWD is defined as:

$$W_{c,\lambda}(\mu, \nu) = \min_{\pi \in \Pi} \int c(\mathbf{x}, \mathbf{y}) d\pi(\mathbf{x}, \mathbf{y}) + \lambda \int \log \left( \frac{d\pi(\mathbf{x}, \mathbf{y})}{d\mu(\mathbf{x}) d\nu(\mathbf{y})} \right) d\pi(\mathbf{x}, \mathbf{y}) \tag{1}$$

where $\Pi = \{\pi(\mathbf{x}, \mathbf{y}) \in \mathcal{P}(\mathcal{X} \times \mathcal{X}) | \int \pi(\mathbf{x}, \cdot) d\mathbf{x} = \nu, \int \pi(\cdot, \mathbf{y}) d\mathbf{y} = \mu\}$.

The Sinkhorn divergence uses auto-correlation terms to reduce the entropic bias introduced by ERWD with respect to the exact Wasserstein distance, canceling it out completely for $\mu = \nu$ (i.e. $S_{c,\lambda}(\mu, \nu) = 0$ for matching $\mu = \nu$). This results in faithful matching between the generator and real distributions. Here, we use the Sinkhorn divergence as defined in [37].

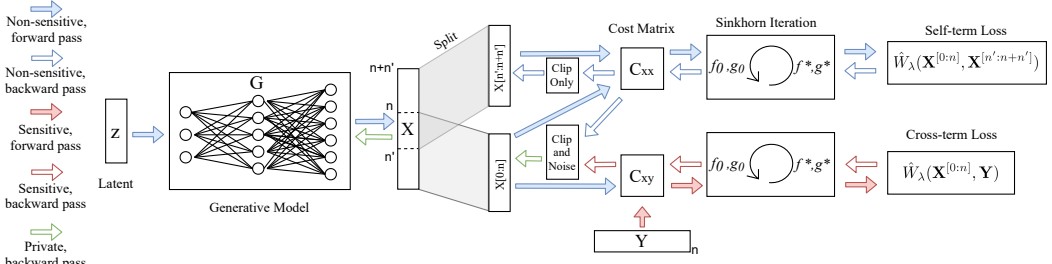

Figure 1: Flow diagram of DP-Sinkhorn for a single training iteration: Batch of generated data is split for the calculation of the cross term and the self term losses. Element-wise differences are captured in the cost matrix. Then, the losses are calculated using the Sinkhorn algorithm. In the backward pass, we impose a privacy barrier behind the generator by clipping and adding noise to the gradients at the generated image level, similar to [13].

**Definition 3.1.** *(Sinkhorn Divergence) The Sinkhorn divergence between measures $\mu$ and $\nu$ is defined as:*

$$S_{c,\lambda}(\mu,\nu) = 2W_{c,\lambda}(\mu,\nu) - W_{c,\lambda}(\mu,\mu) - W_{c,\lambda}(\nu,\nu) \tag{2}$$

### 3.3 Differential Privacy

The current gold standard for measuring the privacy risk of data-releasing programs is the notion of differential privacy (DP) [5]. Informally, DP measures to what degree a program's output can deviate between adjacent input datasets $d$ and $d'$—sets differing by one entry. For a user contributing their data, this translates to a guarantee on how much an adversary could learn about them from observing the program's output. Here, we are learning a generative model of images, while conditioning on available semantic labels. Hence, we are interested in the domain of image-and-label datasets where each image and its label constitute an entry.

A well-studied formulation of privacy, which allows tight composition of multiple queries and can be easily converted to standard definitions of DP, is provided by Rényi Differential Privacy (RDP) [38]:

**Definition 3.2.** *(Rényi Differential Privacy) A randomized mechanism $\mathcal{M} : \mathcal{D} \to \mathcal{R}$ with domain $\mathcal{D}$ and range $\mathcal{R}$ satisfies $(\alpha, \epsilon)$-RDP if for any adjacent $d, d' \in \mathcal{D}$:*

$$D_\alpha(\mathcal{M}(d)|\mathcal{M}(d')) \le \epsilon, \tag{3}$$

*where $D_\alpha$ is the Rényi divergence of order $\alpha$. Also, any $\mathcal{M}$ that satisfies $(\alpha, \epsilon)$-RDP also satisfies $(\epsilon + \frac{\log 1/\delta}{\alpha-1}, \delta)$-DP*

Here, $\mathcal{M}$ is a DP-learning algorithm, $d$ a training set, and $\mathcal{M}(d)$ a generator trained on $d$. The randomized mechanism can often be dissected into a deterministic function of the dataset $\mathcal{Q} : \mathcal{D} \to \mathcal{R}$, the query, and a noise injecting random function $\mathcal{M}' : \mathcal{R} \to \mathcal{R}$, the privacy mechanism, such that $\mathcal{M}(d) = \mathcal{M}' \circ \mathcal{Q}(d)$. The sensitivity of a query $S(\mathcal{Q})$ is a property that represents the maximum magnitude of change between outputs of the query when applied to adjacent datasets. For a query output $q = \mathcal{Q}(d)$ with sensitivity $S(\mathcal{Q})$ and standard deviation of Gaussian noise $\sigma$, the Gaussian mechanism $M(q) = q + z$, $z \sim N(0, \sigma^2)$ satisfies $(\alpha, \alpha S(\mathcal{Q})^2/(2\sigma^2))$-RDP [38]. Subsampling the dataset into batches also improves privacy. The effect of subsampling on the Gaussian mechanism under RDP has been studied in [39, 40, 41]. Furthermore, privacy analysis of a gradient-based learning algorithm entails accounting for the privacy cost of single queries (possibly with subsampling), summing up the privacy cost across all queries (i.e. training iterations in our case), and then choosing the best $\alpha$. A more thorough discussion of DP can be found in the Appendix.

## 4 Differentially Private Sinkhorn

We propose DP-Sinkhorn (Fig. 1), an OT-based method to learn differentially private generative models that avoids the training instability issues of GANs. In this section, we first provide an overview of DP-Sinkhorn, followed by our novel semi-debiased Sinkhorn loss function. We then analyze the privacy protection of DP-Sinkhorn, and discuss some design considerations.

### 4.1 Overview of DP-Sinkhorn

DP-Sinkhorn aims to stably and robustly train generative models on high dimensional data (e.g. images) while preserving the privacy of training data. As discussed in Sec. 2, current state-of-the-art methods in privacy-preserving data generation are reliant on adversarial training schemes that are not robust, unstable, and complicated to train. DP-Sinkhorn leverages advancements in OT-based generative learning to do away with the adversarial training scheme. Specifically, training a generative model with DP-Sinkhorn is a straightforward end-to-end iterative loss minimization process. In each iteration, data produced by the generator are split according to the debiasing ratio into a "cross" group and a "debiasing" group. Empirical OT distances are calculated between the "cross" group and the real data, and between the "debiasing" group and the "cross group". Gradients of the OT distances with respect to the generated data are calculated and backpropagated to the generator. Privacy protection is enforced by clipping and adding noise to the gradients of the "cross" group during backpropagation.

**Algorithm 1** *DP-Sinkhorn*

$L$ is number of categories. $\mathcal{X}$ is sample space. $M$ is size of private data set. $backprop$ is a reverse mode auto-differentiation function that takes 'out', 'in' and 'grad weights' as input and computes the Jacobian vector product $J_{\text{in}}(\text{out}) \cdot$ grad weights. Poisson Sample and $\hat{W}_\lambda$ (via Sinkhorn iterations) are defined in Appendix.

---

**Input:** private data set $d = \{(\mathbf{y}, \text{l}) \in \mathcal{X} \times \{0, ..., L\}\}^M$, sampling ratio $q$, noise scale $\sigma$, clipping coefficient $\Delta$, generator $g_\theta$, learning rate $\alpha$, entropy regularization $\lambda$, debiasing resample fraction $p$, total steps $T$.
**Output:** $\theta$
$n = q * M$, $n' = floor(n * p)$
**for** $t = 1$ **to** $T$ **do**
    Sample $\mathbf{Y} \leftarrow$ Poisson Sample$(d, q)$,
    $\mathbf{Z} \leftarrow (\mathbf{z}_i)_{i=1}^{(n+n')} \overset{i.i.d.}{\sim}$ Unif$(0, 1)$
    $L_x \leftarrow \{\text{l}_i\}_{i=1}^{(n+n')} \overset{i.i.d.}{\sim}$ Unif$(0, ..., L)$
    $\mathbf{X} \leftarrow \{\mathbf{x}_i = g_\theta(\mathbf{z}_i, \text{l}_i)\}_{i=1}^{(n+n')}$
    $\text{grad}_\mathbf{X} \leftarrow \nabla_\mathbf{X} \hat{S}_{c,\lambda,p}(\mathbf{X}, \mathbf{Y})$
    $\text{grad}_{\mathbf{X}[0:n]} \leftarrow clip(\text{grad}_{\mathbf{X}[0:n]}, \Delta) + 2\Delta\sigma\mathcal{N}(\vec{0}, \mathbb{I})$
    $\text{grad}_{\mathbf{X}[n:n+n']} \leftarrow clip(\text{grad}_{\mathbf{X}[n:n+n']}, \Delta)$
    $\text{grad}_\theta \leftarrow backprop(\mathbf{X}, \theta, \text{grad}_\mathbf{X})$
    $\theta \leftarrow \theta - \alpha * Adam(\text{grad}_\theta)$
**end for**

---

### 4.2 Estimating Sinkhorn Divergence with Semi-Debiased Sinkhorn Loss

Sinkhorn divergence, as expressed in Eq. 2, involves integration over the sample space. Empirical estimation of Eq. 2 based on finite samples is required to train a generative model through gradient-based optimization. A solution suggested by previous works [37, 21] would be to replace $\mu$ and $\nu$ with empirical samples from each distribution.

**Definition 4.1.** *The empirical Sinkhorn loss computed over a batch of $n$ generated examples $\mathbf{X}$ and $m$ real examples $\mathbf{Y}$ is defined as [37]:*

$$\hat{S}_{c,\lambda}(\mathbf{X}, \mathbf{Y}) = 2\hat{W}_\lambda(\mathbf{X}, \mathbf{Y}) - \hat{W}_\lambda(\mathbf{X}, \mathbf{X}) - \hat{W}_\lambda(\mathbf{Y}, \mathbf{Y}), \tag{4}$$

*where $\hat{W}_\lambda(\mathbf{A}, \mathbf{B}) = C_{\mathbf{AB}} \odot P_{\mathbf{AB}}^\lambda$.[2] $C_{\mathbf{AB}} \in \mathbb{R}^{+n \times m}$ with $C_{i,j} = c(\mathbf{x}_i, \mathbf{y}_j)$ ($\mathbf{x}_i, \mathbf{y}_j$ are rows of $\mathbf{A}, \mathbf{B}$) is the cost matrix between $\mathbf{A}$ and $\mathbf{B}$, and $P_{\mathbf{AB}}^\lambda$ is the approximate optimal transport plan that empirically minimizes $\hat{W}_\lambda(\mathbf{A}, \mathbf{B})$ computed by the Sinkhorn algorithm.*

However, [42] showed that the gradients of $\hat{S}_{c,\lambda}(\mathbf{X}, \mathbf{Y})$ are biased estimates of the gradients of $S_{c,\lambda}(\mu, \nu)$, computed over the population. Instead, they proposed a loss formulation that produces unbiased gradients using additional independently drawn samples:

**Definition 4.2.** *Following the notations of Def. 4.1, let $\mathbf{X}'$ and $\mathbf{Y}'$ denote a second batch of generated and real examples. The debiased Sinkhorn loss is then defined as [42]:*

$$\hat{S}_{c,\lambda}(\mathbf{X}, \mathbf{Y}, \mathbf{X}', \mathbf{Y}') = 2\hat{W}_\lambda(\mathbf{X}, \mathbf{Y}) - \hat{W}_\lambda(\mathbf{X}, \mathbf{X}') - \hat{W}_\lambda(\mathbf{Y}, \mathbf{Y}'). \tag{5}$$

In comparison with Def. 4.1, Def. 4.2 comes with higher variance (only $\hat{W}_\lambda(\mathbf{X}, \mathbf{Y})$ contributes to variance in Def. 4.1). Unfortunately, privacy constraints in the DP setting prevent us from using very large batch sizes or very long training periods with low learning rates to effectively reduce variance. Hence, the variance incurred from using the unbiased estimator is more difficult to handle in the DP setup. Furthermore, Def. 4.2 draws two batches of real data in every training step, thereby increasing the privacy cost of each step. Nonetheless, Def. 4.2 is an unbiased estimator with better convergence properties. We now discuss how we overcome the above issues in DP-Sinkhorn.

---

[2] $\odot$ is the Hadamard product.

First, we make the observation that the $\hat{W}_\lambda(\mathbf{Y}, \mathbf{Y}')$ term does not contribute to gradients of the generator. Hence, we can omit it from $\hat{S}_{c,\lambda}$. Next, we propose a loss formulation that interpolates between biased and unbiased Sinkhorn divergence. As observed in previous works, it can sometimes be beneficial to control bias-variance trade-offs through mixing biased and unbiased gradient estimators [43]. Instead of completely resampling the generator for $\mathbf{X}'$, we reuse some of the samples in $\mathbf{X}$ when computing $\hat{W}_\lambda(\mathbf{X}, \mathbf{X}')$. This provides better control over the bias-variance trade-off when empirically estimating gradients.

**Definition 4.3.** *(Semi-debiased Sinkhorn loss) For a mixture fraction $p \in [0, 1]$ and natural number $n$, $n' = floor(n \times p)$. Given $n + n'$ generated samples $\mathbf{X} \in \mathcal{X}^{n+n'}$ and $m$ real samples $\mathbf{Y} \in \mathcal{X}^m$, the semi-debiased Sinkhorn loss is defined as:*

$$\hat{S}_{c,\lambda,p}(\mathbf{X}, \mathbf{Y}) = 2\hat{W}_\lambda(\mathbf{X}^{[0:n]}, \mathbf{Y}) - \hat{W}_\lambda(\mathbf{X}^{[0:n]}, \mathbf{X}^{[n':n+n']}), \tag{6}$$

*where $\mathbf{X}^{[a:b]}$ denotes the contiguous rows of $\mathbf{X}$ starting from $a$ and ending with $b - 1$.*

When $p = 1$, Eq. 6 is equal to Eq. 5, whereas when $p = 0$, Eq. 6 recovers Eq. 4 (ignoring the terms in Eqs. 4 and 5 that only depend on data $\mathbf{Y}$ and are irrelevant during training).

Algorithm 1 describes how Eq. 6 is used to train a generative model, while additionally modifying the gradient by adding noise and clipping to implement the privacy mechanism described below. Training of the generator proceeds by computing the gradient of the semi-debiased Sinkhorn loss with respect to $\mathbf{X}$. Please also see the Appendix for more details.

## 4.3 Privacy Protection

Information about real data enters the generator through loss function gradients with respect to the generated images. Let $\mathbf{G} = \nabla_\mathbf{X} \hat{S}_{c,\lambda,p}(\mathbf{X}, \mathbf{Y})$ denote the gradients of the semi-debiased Sinkhorn loss, and let $\mathbf{G}^{[a:b]}$ denote the contiguous rows of $\mathbf{G}$ from $a$ to $b - 1$ inclusive. We modify $\mathbf{G}$ as:

$$\tilde{\mathbf{G}} = \mathbf{G}^{[0:n]} \cdot \min\left(\frac{\Delta}{||\mathbf{G}^{[0:n]}||_2}, 1\right), \quad \tilde{\mathbf{G}}' = \mathbf{G}^{[n:n+n']} \cdot \min\left(\frac{\Delta}{||\mathbf{G}^{[n:n+n']}||_2}, 1\right)$$

$$\hat{\mathbf{G}} = \text{concat}(\tilde{\mathbf{G}} + \gamma, \tilde{\mathbf{G}}'), \quad \text{where } \gamma \sim \mathcal{N}(0, \Delta^2 \sigma^2), \quad \text{concat is applied to the first axis}$$

We observe that $\nabla_{\mathbf{X}^{[n:n+n']}} \hat{W}_\lambda(\mathbf{X}^{[0:n]}, \mathbf{Y}) = 0$, i.e. $\mathbf{G}^{[n:n+n']}$ contains no information about $\mathbf{Y}$. As such, noise does not need to be added to this term, but we apply clipping to $\mathbf{G}^{[n:n+n']}$ to preserve the scale between the magnitudes of the gradients. In addition, the sensitivity of $\tilde{\mathbf{G}}$ is $\max_{\mathbf{Y}, \mathbf{Y}'} ||\tilde{\mathbf{G}}(\mathbf{Y}) - \tilde{\mathbf{G}}(\mathbf{Y}')||_2 \leq 2\Delta$. The following theorem states the privacy guarantee of DP-Sinkhorn's gradient updates, with proofs in the appendix:

**Theorem 4.1.** *For clipping constant $\Delta$ and noise vector $\gamma \sim \mathcal{N}(0, \Delta^2 \sigma^2)$, releasing $\hat{\mathbf{G}}$ satisfies $(\alpha, 2\alpha/\sigma^2)$-RDP.*

We use the RDP accountant with Poisson subsampling proposed in [41] for privacy composition across updates. Note that the batch size of $\mathbf{X}$ is kept fixed, while the batch size of $\mathbf{Y}$ follows a binomial distribution due to Poisson subsampling.

## 4.4 Design Considerations

**Advantages of primal form OT** When compared to WGAN [33], learning with primal form OT (such as Sinkhorn divergence) has distinct differences. While both are approximations to the exact Wasserstein distance, the source of the approximation error differs. WGAN's source of error lies in the sub-optimality of the dual potential function. Since this potential function is parameterized by an adversarially trained deep neural network, it enjoys neither convergence guarantees nor feasibility guarantees. Furthermore, the adversarial training scheme can cause the discriminator and generator to change abruptly every iteration to counter the strategy of the other player from the previous iteration [44], resulting in non-convergence. These challenges are exacerbated in the DP setting. In contrast, the suboptimality of the transport plan when computing Sinkhorn divergence can be controlled by using enough iterations, and the bias introduced by entropic regularization can be controlled by using small $\lambda$ values. Training with the Sinkhorn divergence does not involve any adversarial training, converges more stably, and reaps the benefits of OT metrics at covering modes.

Table 1: Comparison of DP image generation results on MNIST and Fashion-MNIST at $(\epsilon, \delta) = (10, 10^{-5})$-DP. Results for other methods (G-PATE [27], DP-MERF AE [29], DP-CGAN [24], GS-WGAN [13]) are from [13], except Datalens [14]. Results are averaged over 5 runs of synthetic dataset generation and classifier training.

| Method | DP-$\epsilon$ | MNIST | | | | Fashion-MNIST | | | |
| | | FID | Acc (%) | | | FID | Acc (%) | | |
| | | | Log Reg | MLP | CNN | | Log Reg | MLP | CNN |
|---|---|---|---|---|---|---|---|---|---|
| Real data | $\infty$ | 1.6 | 92.2 | 97.5 | 99.3 | 2.5 | 84.5 | 88.2 | 90.8 |
| Non-priv Sinkhorn ($m=1$) | $\infty$ | 54.2 | 89.0 | 89.0 | 91.0 | 65.8 | 78.4 | 79.1 | 73.9 |
| Non-priv Sinkhorn ($m=3$) | $\infty$ | 43.4 | 87.7 | 87.3 | 90.6 | 63.8 | 78.4 | 78.4 | 73.3 |
| G-PATE | 10 | 177.2 | 26 | 25 | 51/80.9[3] | 205.8 | 42 | 30 | 50/69.3[3] |
| DP-CGAN | 10 | 179.2 | 60 | 60 | 63 | 243.8 | 51 | 50 | 46 |
| DP-MERF AE | 10 | 161.1 | 54 | 55 | 68 | 213.6 | 50 | 56 | 62 |
| DataLens | 10 | 173.5 | N/A | N/A | 80.66 | 167.7 | N/A | N/A | 70.61 |
| GS-WGAN | 10 | 61.3 | 79 | 79 | 80 | 131.3 | 68 | 65 | 65 |
| DP-Sinkhorn ($m=1$) | 10 | 61.2 | **79.5** | **80.2** | **83.2** | 145.1 | **73.0** | **72.8** | **70.9** |
| DP-Sinkhorn ($m=3$) | 10 | **55.56** | 79.1 | 79.2 | 79.1 | **129.4** | 70.2 | 70.2 | 68.9 |

**Cost function**    The choice of the element-wise cost $c(\mathbf{x}, \mathbf{y})$ influences the type of images produced by the generator. We consider a mixture between pixel-wise $L_1$ and squared $L_2$ losses. $L_2$ loss has smooth gradients that scale with the difference in pixel value, whereas the gradient of $L_1$ loss is constant in magnitude for each pixel that differs. Therefore, while $L_2$ loss can quickly rein in outlier pixel values, $L_1$ loss can encourage generated image pixels to closely match those of the real image, promoting sharpness. We define the element-wise cost function as $c_m(\mathbf{x}, \mathbf{y}) = L_2(\mathbf{x}, \mathbf{y})^2 + m\,L_1(\mathbf{x}, \mathbf{y})$, where $L_2(\mathbf{x}, \mathbf{y}) = ||\mathbf{x} - \mathbf{y}||_2$, $L_1(\mathbf{x}, \mathbf{y}) = |\mathbf{x} - \mathbf{y}|$ and $m$ is a scalar mixture weight. Class conditioning is also achieved through the cost function by concatenating a one-hot class embedding to both the generated images and real images, similar to the approach used in [42]. Intuitively, this works by increasing the cost between image pairs of different classes, hence shifting the weight of the transport plan ($P_\lambda^*$ in Eq. 4) towards class-matched pairs.

# 5    Experiments

We conduct experiments on differentially private conditional image synthesis, since our focus is on generating high-dimensional data with privacy protection. We evaluate our method on both visual quality and data utility for downstream classification tasks. Additional experiments and analyses of the proposed semi-debiased Sinkhorn loss can be found in the Appendix. Code will be released through the project page[4].

## 5.1    Experimental Setup

**Datasets**    We use 3 image datasets: MNIST [45], Fashion-MNIST [46], and CelebA [47] down-sampled to 32x32 pixels. For MNIST and Fashion-MNIST, generation is conditioned on regular class labels; for CelebA we condition on gender.

**Metrics**    In all experiments, we compute metrics against a synthetic dataset of 60k image-label pairs sampled from the model. For a quantitative measure of visual quality, we report FID [48]. To measure the utility of generated data, we assess the class prediction accuracy of classifiers trained with synthetic data on the real test sets. We consider logistic regression, MLP, and CNN classifiers.

**Architectures & Hyperparameters**    We implement DP-Sinkhorn with two generator architectures. We adopt a four layer, convolutional architecture from DCGAN [49] for MNIST and Fashion-MNIST experiments, and a twelve layer residual architecture from BigGAN [15] for CelebA experiments. Class conditioning is achieved by providing a one-hot encoding of the label to the generator, and concatenating the one-hot encoding to images when calculating the element-wise cost. We set $\lambda$=0.05 for MNIST and Fashion-MNIST experiments, and $\lambda$=5 for CelebA experiments. Complete implementation details can be found in the Appendix.

---

[3] The G-PATE [27] authors report much more accurate classification results than reported in [13]. The visual quality of samples in both papers is roughly the same.

[4] https://nv-tlabs.github.io/DP-Sinkhorn

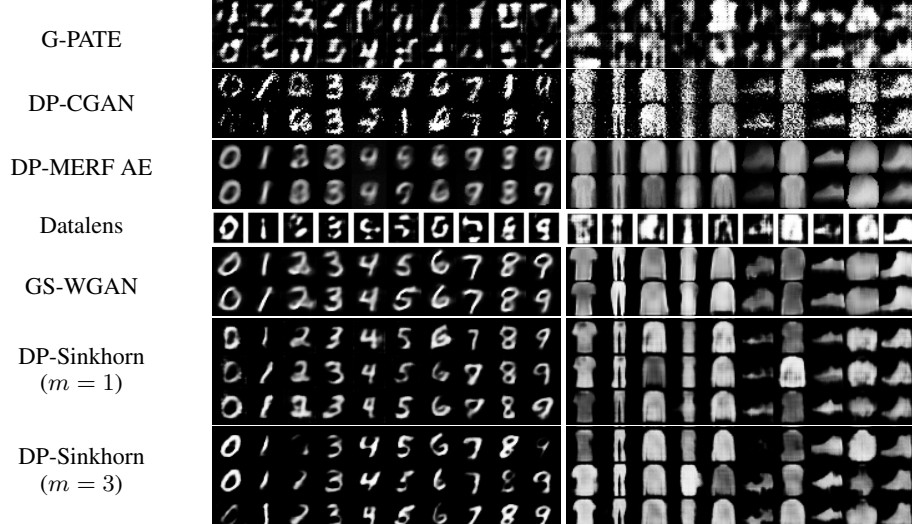

Figure 2: Images generated at $(10, 10^{-5})$-DP for MNIST and Fashion-MNIST by various methods. Datalens images obtained from [14]; images of other methods obtained from [13].

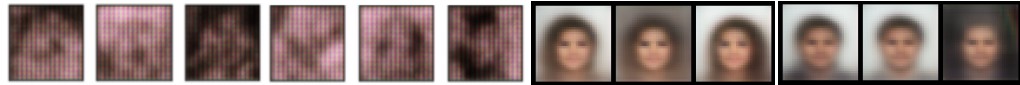

Figure 3: Images generated on CelebA by Datalens (Left) and DP-Sinkhorn (Right). Datalens images obtained from [14].

**Privacy Implementation**   Our models are implemented in PyTorch. We implement the gradient sanitization mechanism by registering a backward hook to the generator output. MNIST and Fashion-MNIST experiments target $(10, 10^{-5})$-DP while CelebA experiments target $(10, 10^{-6})$-DP. Details are in the Appendix.

## 5.2   Experimental Results on Standard Benchmarks

In Table 1, we compare the performance of two DP-Sinkhorn variants with other methods on MNIST and Fashion-MNIST. We use $p$=0.2 for the semi-debiased loss, which was determined through grid search. The two variants use different weights ($m$=1 and $m$=3) for the $L_1$ loss in the cost function. Given the same privacy budget, DP-Sinkhorn with $m$=1 generates more informative examples than previous methods, as demonstrated by the higher accuracy achieved by the downstream classifier. On the more visually complex Fashion-MNIST, DP-Sinkhorn's lead is especially pronounced, beating previous state-of-the-art results by a significant margin. DP-Sinkhorn with $m$=3 achieves lower FID than all baselines, while still maintaining downstream accuracy similar to GS-WGAN. We hypothesize that giving more weight to the $L_1$ loss improves FID because $L_1$ is more sensitive to small differences in pixel values, thereby encouraging sharper edges. Images generated by DP-Sinkhorn are visualized in Fig. 2. DP-Sinkhorn produces more visual diversity within each class compared to the baselines, which likely benefits DP-Sinkhorn's downstream classification performance.

**Robustness**   We evaluate the training stability of DP-Sinkhorn ($m = 1$, $p = 0.2$) with different learning rates and two optimizers (Adam [50] and SGD) on MNIST. We perform the same parameter sweep on GS-WGAN for comparison[5], as it is the strongest baseline we are comparing to. Results are illustrated in Fig. 4a. We find that DP-Sinkhorn reliably converges for sufficiently small learning rates, and it is not sensitive to the choice of optimizer. In contrast, GS-WGAN, relying on adversarial training, suffers from non-convergence for learning rates too big or too small, and is very sensitive to the choice of optimizer. Exact numbers are reported in the Appendix.

**Privacy Utility Trade-off**   Stronger privacy protection can be attained by training DP-Sinkhorn for fewer iterations at the cost of utility and image quality. We evaluate the performance of DP-Sinkhorn at various privacy budgets and contrast it to GS-WGAN (Fig. 4b). DP-Sinkhorn shows strong performance among a wide range of privacy budgets, and provides good downstream utility

---

[5]https://github.com/DingfanChen/GS-WGAN (MIT License) [13]

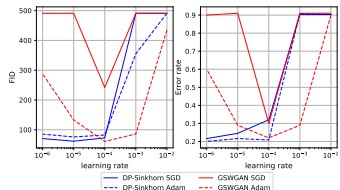 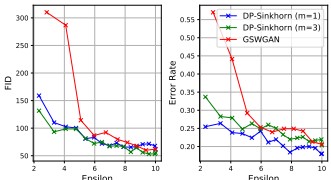 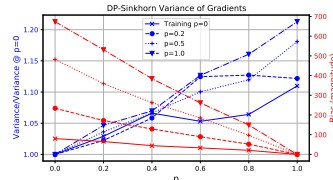

(a) Comparing hyperparameter sensitivity of DP-Sinkhorn to GS-WGAN on MNIST. Error rate is calculated as $1 - $ Accuracy.

(b) FID and utility of DP-Sinkhorn ($m=1$ and $m=3$), and GS-WGAN at various $\epsilon$ on MNIST.

(c) Bias-variance trade-off of the gradient estimator over semi-debiasing parameter $p$.

Figure 4: Analyzing hyperparameter choices in DP-Sinkhorn.

even at a small privacy budget of $\epsilon = 2.33$, significantly outperforming GS-WGAN. Note that we found GS-WGAN to require significantly more memory than DP-Sinkhorn, since it uses multiple discriminators for different parts of the data. In our experiments, DP-Sinkhorn can fit comfortably on an 11GB GPU, while GS-WGAN requires 24GB of GPU memory. Hence, DP-Sinkhorn is arguably more scalable to very large datsets.

**Analysis of Semi-debiased Sinkhorn Loss**  To study why our novel semi-debiased Sinkhorn loss outperforms both fully-debiased and fully-biased Sinkhorn losses, we evaluate bias and variance of the semi-debiased Sinkhorn loss-based gradient estimator $G_p = \nabla_\theta \hat{S}(\mathbf{X}(\theta), \mathbf{Y})$. We sample generator gradients with respect to the semi-debiased Sinkhorn loss with different $p$ and plot bias and variance (Fig. 4c). Each line represents a generator trained with a different $p$ on MNIST. For each value of $p$, we compute $G_p$ on three hundred batches of real and generated data to obtain its average and sample variance. Since $G_1$ is unbiased, we use it as the ground truth when computing bias. Variances of each model's gradients are normalized with respect to variance of $G_0$.

We observe two prominent trends from this graph. First, as we increase $p$, bias decreases and variance increases. This requires us to find a balance in the trade-off between bias and variance. Second, we see flatter curves for generators trained

Table 2: Ablating loss functions, debiasing, and gradient perturbation mechanism on MNIST.

| Image Gradient Perturbation | Loss | Debiasing | FID | Acc (%) | |
|---|---|---|---|---|---|
| | | | | MLP | CNN |
| No | $L_2$ | No | 218.6 | 79.9 | 80.7 |
| Yes | $L_2$ | No | 124.3 | 82.0 | 80.8 |
| Yes | $L_1$ | No | 73.9 | 68.4 | 65.7 |
| Yes | $L_1+L_2$ | No | 88.6 | 76.6 | 76.1 |
| Yes | $L_1+L_2$ | Full | 98.0 | 63.0 | 60.5 |
| Yes | $L_1+L_2$ | Semi | 61.2 | 80.2 | 83.2 |

Table 3: DP image generation results on down-sampled CelebA. We include results from [14] for context, but note that their experiment uses a 64x64 resolution and a larger $\delta$ of $10^{-5}$.

| Method | $(\epsilon, 10^{-6})$-DP | FID | Acc (%) | |
|---|---|---|---|---|
| | | | MLP | CNN |
| Real data | $\infty$ | 1.1 | 91.9 | 95.0 |
| Sinkhorn | $\infty$ | 129.5 | 80.8 | 82.2 |
| DP-Sinkhorn | 10 | 168.4 | 76.2 | 75.8 |
| DataLens [14] | $(10, 10^{-5})$ | 320.8 | N/A | 72.9 |

with smaller $p$. As $p$ affects bias and variance through changing the number of resampled generated images, we can deduce that training with smaller $p$ likely results in greater similarity between generated images, which improves consistency across generated images at the cost of diversity. That is, if the generator is mode collapsed, $p$ would have no effect on the bias-variance trade-off, as resampling the latent variables would produce the same images. While previous works [42] found fully-debiased ($p = 1$) Sinkhorn loss to provide higher performance, we find a small amount of debiasing ($p = 0.2$) to perform best. Our hypothesis is that because training in a privacy-preserving manner is restrictive in batch size and number of iterations, the increased variance of the fully-debiased loss is more detrimental. In particular, in the DP setting we cannot simply increase batch sizes or train with more iterations and lower learning rates to counteract high loss variances, as this would incur increased privacy costs. In contrast, our novel semi-debiasing provides control over the trade-off between consistent low-variance gradients and less biased objectives. This also demonstrates how training in the DP setting differs from the non-private setting, hence requiring new ideas and tailored methods.

**Ablations**  We study the impact of perturbing image vs. parameter gradients, design of element-wise cost function, and debiasing on performance in the MNIST benchmark. We start with the simplest model, using parameter gradient perturbation, $L_2$ loss and no debiasing, and incrementally add components. We use $m=1$ when adding $L_1$ loss, and $p=0.2$ when adding semi-debiasing. The

clipping bound $\Delta$ is tuned separately for the variant with parameter gradient perturbation, while the other hyperparameters are kept fixed. In Table 2, we see that DP-Sinkhorn with parameter gradients is already competitive in downstream accuracy, but has poor FID in comparison to using image gradients. We observe that DP-Sinkhorn with $L_2$ loss yields good downsteam task performance, but has higher FID than the $L_1$ loss variant. Mixing $L_1$ and $L_2$ loss strikes a balance between better FID and downstream accuracy. We also observe that using a fully debiased gradient estimator is detrimental to performance, which we postulate is due to its high variance. The semi-debiased variant performs better than both the biased and the debiased variants.

### 5.3 Experimental Results on CelebA

We also evaluate DP-Sinkhorn on downsampled CelebA. We evaluate whether DP-Sinkhorn is able to synthesize RGB images that are informative for downstream classification. Despite its simplicity, DP-Sinkhorn generates informative images for gender classification, as seen in Tab. 3 (uninformative images would correspond to a $\approx 50\%$ classification ratio). Qualitatively, Fig. 3 shows that DP-Sinkhorn can learn meaningful representations of each semantic class (male and female) and produces some in-class variations, while avoiding details that could uniquely identify individuals. Concurrent to our work, Datalens [14] was also applied to gender-conditioned generation of CelebA images, albeit with a different image resolution than ours. Images generated by DP-Sinkhorn clearly resemble faces, while those generated by Datalens are blurrier. We also attempted to train GS-WGAN on CelebA, but couldn't obtain meaningful results using the default hyper-parameters.

## 6 Conclusions

We propose DP-Sinkhorn, a novel optimal transport-based differentially private generative model. Our approach minimizes a new semi-debiased Sinkhorn loss in a differentially private manner. It does not require any adversarial techniques that are challenging to optimize. Consequently, DP-Sinkhorn is easy to train, which we hope will help its adoption in practice. We experimentally demonstrate superior performance compared to the previous state-of-the-art both in terms of image quality and on standard image classification benchmarks using data generated under DP. Our model is applicable for varying privacy budgets and is capable of synthesizing informative RGB images in a differentially private way without using additional public data. We conclude that robust models such as ours are a promising direction for differentially private generative modeling.

**Limitations and Future Work** Our main experiments only used simple pixel-wise $L_1$- and $L_2$-losses as cost function, yet achieve better performance than GAN-based methods. This suggests that in the DP setting complexity in model and objective are not necessarily beneficial. Nonetheless, limited image quality is the main challenge in DP generative modeling and future work includes designing more expressive generator networks that can further improve synthesis quality, while satisfying differential privacy. To this end, kernel-based cost functions may provide better performance on suitable datasets. Our experiments were performed on widely-used image benchmarks for differentially private generative learning. Future works may extend our method to other data types and domains. In particular, since privacy is an important consideration for medical data, applying DP-Sinkhorn to medical datasets (such as those used in [51]) could be of high practical interest.

**Broader Impact** Our work improves the state-of-the-art in privacy-preserving generative modeling. Such advances promise significant benefits to the machine learning community, by allowing sensitive data to be shared more broadly via privacy-preserving generative models. We believe the strong performance and robustness of DP-Sinkhorn will facilitate its adoption by practitioners. Although DP-Sinkhorn provides privacy protection in generative learning, information about individuals cannot be eliminated entirely, as no useful model can be learned under $(0, 0)$-DP. This should be communicated clearly to dataset participants. We recognize that classifiers learned with DP can potentially underperform for minority members within the dataset [52, 53, 54], which may also be the case for classifiers trained on data produced by DP-Sinkhorn. Addressing these types of imbalances is an active area of research [55, 56, 57, 58].

## Acknowledgments and Disclosure of Funding

This work was funded by NVIDIA. Tianshi Cao and Alex Bie acknowledge additional revenue from Vector Scholarships in Artificial Intelligence, which are not in direct support of this work.

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
