# A    Optimal Transport via the Sinkhorn Divergence

In addition to the notations defined in Sec. 3.1, we denote the Dirac delta distribution at $\mathbf{x} \in \mathcal{X}$ as $\delta_{\mathbf{x}}$, and the standard $n$-simplex as $\mathcal{S}^n$.

Recall from Sec. 3.2 that, given a positive cost function $c : \mathcal{X} \times \mathcal{X} \mapsto \mathbb{R}^+$ and $\lambda \geq 0$, the Entropy Regularized Wasserstein Distance is defined as:

$$W_{c,\lambda}(\mu, \nu) = \min_{\pi \in \Pi} \int c(\mathbf{x}, \mathbf{y}) d\pi(\mathbf{x}, \mathbf{y}) + \lambda \int \log\left(\frac{d\pi(\mathbf{x}, \mathbf{y})}{d\mu(\mathbf{x})d\nu(\mathbf{y})}\right) d\pi(\mathbf{x}, \mathbf{y}) \tag{7}$$

where $\Pi = \left\{ \pi(\mathbf{x}, \mathbf{y}) \in \mathcal{P}(\mathcal{X} \times \mathcal{X}) | \int \pi(\mathbf{x}, \cdot)d\mathbf{x} = \nu, \int \pi(\cdot, \mathbf{y})d\mathbf{y} = \mu \right\}$.

We use the Sinkhorn divergence, as defined in [37].

**Definition A.1.** *(Sinkhorn Loss) The Sinkhorn loss between measures $\mu$ and $\nu$ is defined as:*

$$S_{c,\lambda}(\mu, \nu) = 2W_{c,\lambda}(\mu, \nu) - W_{c,\lambda}(\mu, \mu) - W_{c,\lambda}(\nu, \nu) \tag{8}$$

For modeling data-defined distributions, as in our situation, an empirical version can be defined, too. Note that we use a slightly different notation compared to the main text, because it is more convenient to deal with empirical distributions rather than samples when relating to the dual formulation later on.

**Definition A.2.** *(Empirical Sinkhorn loss) The empirical Sinkhorn loss computed over a batch of $N$ generated examples and $M$ real examples is defined as:*

$$\hat{S}_{c,\lambda}(\hat{\mu}, \hat{\nu}) = 2C_{\mathbf{XY}} \odot P^*_{\lambda, \mathbf{X}, \mathbf{Y}} - C_{\mathbf{XX}} \odot P^*_{\lambda, \mathbf{X}, \mathbf{X}} - C_{\mathbf{YY}} \odot P^*_{\lambda, \mathbf{Y}, \mathbf{Y}} \tag{9}$$

*where $\hat{\mu} = \frac{1}{N}\sum_{i=1}^{N} \delta_{\mathbf{x}_i}$, and $\hat{\nu} = \frac{1}{M}\sum_{j=1}^{M} \delta_{\mathbf{y}_j}$. For two samples $\mathbf{A} \in \mathcal{X}^N$ and $\mathbf{B} \in \mathcal{X}^M$, $C_{\mathbf{A},\mathbf{B}}$ is the cost matrix between $\mathbf{A}$ and $\mathbf{B}$, and $P^*_{\lambda, \mathbf{A}, \mathbf{B}}$ is an approximate optimal transport plan that minimizes Eq. 7 computed over $\mathbf{A}$ and $\mathbf{B}$.*

$P^*_{\lambda}$ is arrived at by iterating the dual potentials: [23] and [37] have shown the following dual formulation for the discretized version of $\hat{W}_{c,\lambda}$:

$$\hat{W}_{c,\lambda}(\hat{\mu}, \hat{\nu}) = \max_{f,g \in \mathcal{S}^N \times \mathcal{S}^M} \langle \hat{\mu}, f \rangle + \langle \hat{\nu}, g \rangle - \lambda \langle \hat{\mu} \otimes \hat{\nu}, \exp(\frac{1}{\lambda}(f \oplus g - C_{\hat{\mu},\hat{\nu}})) - 1 \rangle, \tag{10}$$

where $\otimes$ denotes the product measure and $\oplus$ denotes the "outer sum" such that the output is a matrix of the sums of pairs of elements from each vector. $C_{\hat{\mu},\hat{\nu}}$ is the cost matrix between each element of $\mathbf{x}$ and $\mathbf{y}$ who are distributed according to $\hat{\mu}$ and $\hat{\nu}$, $C_{ij} = c(\mathbf{x}_i, \mathbf{y}_j)$. Then, the optimal transport plan $P^*_{\lambda}$ relates to the dual potentials by $P^*_{\lambda} = \exp(\frac{1}{\lambda}(f \oplus g - C_{\hat{\mu},\hat{\nu}}))(\hat{\mu} \otimes \hat{\nu})$. Thus, once we find the optimal $f$ and $g$, we can obtain $P^*_{\lambda}$ through this primal-dual relationship. We also know the first-order optimal conditions for $f$ and $g$ through the Karush-Kuhn-Tucker theorem:

$$f_i = -\lambda \log \sum_{j=1}^{M} \exp(\log(\hat{\nu}_j) + \frac{1}{\lambda}g_j - \frac{1}{\lambda}c(\mathbf{x}_i, \mathbf{y}_j)) \quad g_j = -\lambda \log \sum_{i=1}^{N} \exp(\log(\hat{\mu}_i) + \frac{1}{\lambda}f_i - \frac{1}{\lambda}c(\mathbf{x}_i, \mathbf{y}_j))$$

$$\tag{11}$$

To optimize $f$ and $g$, it suffices to apply the Sinkhorn algorithm [23], see Algorithm 3. Readers can refer to [59] for further details.

# B    Differential Privacy

As discussed in Sec. 3.3, differential privacy is the current gold standard for measuring the privacy risk of data releasing programs. It is defined as follows [5]:

**Definition B.1.** *(Differential Privacy) A randomized mechanism $\mathcal{M} : \mathcal{D} \to \mathcal{R}$ with domain $\mathcal{D}$ and range $\mathcal{R}$ satisfies $(\varepsilon, \delta)$-DP if for any two adjacent inputs $d, d' \in \mathcal{D}$ differing by at most one entry, and for any subset of outputs $S \subseteq \mathcal{R}$ it holds that*

$$\mathbf{Pr}\left[\mathcal{M}(d) \in S\right] \leq e^{\varepsilon}\mathbf{Pr}\left[\mathcal{M}(d') \in S\right] + \delta. \tag{12}$$

**Gradient perturbation**: For a parametric function $f_\theta(\mathbf{x})$ parameterized by $\theta$ and loss function $L(f_\theta(\mathbf{x}), \mathbf{y})$, usual mini-batched first-order optimizers update $\theta$ using gradients $\mathbf{g}_t = \frac{1}{N} \sum_{i=1}^{N} \nabla_\theta L(f_\theta(\mathbf{x}_i), \mathbf{y}_i)$. Under gradient perturbation, the gradient $\mathbf{g}_t$ is first clipped in $L_2$ norm by constant $\Delta$, and then noise sampled from $\mathcal{N}(0, \sigma^2 \mathbb{I})$ is added. Since differential privacy is closed under *post-processing*—releasing any transformation of the output of an $(\varepsilon, \delta)$-DP mechanism is still $(\varepsilon, \delta)$-DP [6]—the parameters $\theta$ are also differentially private. The relation between $(\varepsilon, \delta)$ and the perturbation parameters $\Delta$ and $\sigma$ is provided by the following theorem:

**Theorem B.1.** *For $c^2 > 2 \log(1.25/\delta)$, Gaussian mechanism with $\sigma \geq c\Delta/\varepsilon$ satisfies $(\varepsilon, \delta)$ differential privacy. [6]*

**Subsampling**: In stochastic gradient descent (SGD) and related methods, randomly drawn batches of data are used in each training step instead of the full dataset. This subsampling of the dataset can provide amplification of privacy protection since the privacy of any record that is not in the batch is automatically protected. The ratio of sample size to population size (number of training data) is the sub-sampling ratio, commonly referred to as $q$. Smaller $q$ results in less privacy expenditure per query. Privacy bounds for various subsampling methods have been extensively studied and applied [5, 39, 40, 41].

**Composition**: SGD requires the computation of the gradient to be repeated every iteration. The repeated application of privacy mechanisms on the same dataset is analyzed through *composition*. Composition of the Gaussian mechanism has been first analyzed by [24] through the moments accountant method.

We utilize the often used Rényi Differential Privacy [38] (RDP), which is defined through the Rényi divergence between mechanism outputs on adjacent datasets:

**Definition B.2.** *(Rényi Differential Privacy) A randomized mechanism $\mathcal{M} : \mathcal{D} \to \mathcal{R}$ with domain $\mathcal{D}$ and range $\mathcal{R}$ satisfies $(\alpha, \varepsilon)$-RDP if for any adjacent $d, d' \in \mathcal{D}$ it holds that*

$$D_\alpha(\mathcal{M}(d) | \mathcal{M}(d')) \leq \varepsilon, \tag{13}$$

*where $D_\alpha$ is the Rényi divergence of order $\alpha$. Also, any $\mathcal{M}$ that satisfies $(\alpha, \varepsilon)$-RDP also satisfies $(\varepsilon + \frac{\log 1/\delta}{\alpha - 1}, \delta)$-DP.*

As discussed in the main text, RDP is a well-studied formulation of privacy that allows tight composition of multiple queries—training iterations in our case—and can be easily converted to standard definitions of DP with definition B.2. Recall that for sensitivity $S$ and standard deviation of Gaussian noise $\sigma$, the Gaussian mechanism satisfies $(\alpha, \alpha S^2/(2\sigma^2))$-RDP [38]. Privacy analysis of a gradient-based learning algorithm entails accounting for the privacy cost of single queries, which corresponds to training iterations in our case, possibly with subsampling due to mini-batched training. The total privacy cost is obtained by summing up the privacy cost across all queries or training steps, and then choosing the best $\alpha$.

For completeness, the Rényi divergence is defined as: $D_\alpha(P|Q) = \frac{1}{\alpha} \log \mathbb{E}_{\mathbf{x} \in Q} \left[ \frac{P(\mathbf{x})}{Q(\mathbf{x})} \right]^\alpha$.

## B.1 Proof of Theorem 4.1

Recalling definitions from the main text:

$$\tilde{\mathbf{G}} = \mathbf{G}^{[0:n]} \cdot \min\left(\frac{\Delta}{||\mathbf{G}^{[0:n]}||_2}, 1\right), \quad \tilde{\mathbf{G}}' = \mathbf{G}^{[n:n+n']} \cdot \min\left(\frac{\Delta}{||\mathbf{G}^{[n:n+n']}||_2}, 1\right)$$

$$\hat{\mathbf{G}} = \text{concat}(\tilde{\mathbf{G}} + \gamma, \tilde{\mathbf{G}}'), \quad \text{where } \gamma \sim \mathcal{N}(0, \Delta^2 \sigma^2), \quad \text{concat is applied to the first axis.}$$

**Theorem.** *For clipping constant $\Delta$ and noise vector $\gamma \sim \mathcal{N}(0, \Delta^2 \sigma^2)$, releasing $\hat{\mathbf{G}}$ satisfies $(\alpha, 2\alpha/\sigma^2)$-RDP.*

*Proof.* The proof relies on three simple steps: *(i)* Deriving the privacy protection for releasing $\tilde{\mathbf{G}} + \gamma$, following standard methods. *(ii)* Showing that $\tilde{\mathbf{G}}'$ carries no information about the data $\mathbf{Y}$. *(iii)* Showing that the concatenated gradients $\hat{\mathbf{G}}$ enjoy the same privacy protection as $\tilde{\mathbf{G}} + \gamma$.

*(i)* Recall from main text $\tilde{\mathbf{G}} = \mathbf{G}^{[0:n]} \cdot \min\left(\frac{\Delta}{||\mathbf{G}^{[0:n]}||_2}, 1\right)$. Clearly, $\max ||\tilde{\mathbf{G}}||_2 \leq \Delta$. Hence, the sensitivity of $\tilde{\mathbf{G}}$ is $\max_{\mathbf{Y},\mathbf{Y}'} ||\tilde{\mathbf{G}}(\mathbf{Y}) - \tilde{\mathbf{G}}(\mathbf{Y}')||_2 \leq 2\Delta$. Therefore, by standard arguments [38], releasing $\tilde{\mathbf{G}} + \gamma$ satisfies $(\alpha, 2\alpha/\sigma^2)$-RDP.

*(ii)* Further, note that $\mathbf{Y}$ is only involved in calculating $\hat{W}_\lambda(\mathbf{X}^{[0:n]}, \mathbf{Y})$. That is, $\hat{W}_\lambda(\mathbf{X}^{[0:n]}, \mathbf{X}^{[n':n+n']})$ contains no information about $\mathbf{Y}$. We also have that $\nabla_{\mathbf{X}^{[n:n+n']'}} \hat{W}_\lambda(\mathbf{X}^{[0:n]}, \mathbf{Y}) = 0$. Therefore, we can show that $\mathbf{G}^{[n:n+n']}$ contains no information about $\mathbf{Y}$:

$$
\begin{aligned}
\mathbf{G}^{[n:n+n']} &= \nabla_{\mathbf{X}^{[n:n+n']'}}(2\hat{W}_\lambda(\mathbf{X}^{[0:n]}, \mathbf{Y}) - \hat{W}_\lambda(\mathbf{X}^{[0:n]}, \mathbf{X}^{[n':n+n']})) \\
&= 2\nabla_{\mathbf{X}^{[n:n+n']'}}\hat{W}_\lambda(\mathbf{X}^{[0:n]}, \mathbf{Y}) - \nabla_{\mathbf{X}^{[n:n+n']'}}\hat{W}_\lambda(\mathbf{X}^{[0:n]}, \mathbf{X}^{[n':n+n']}) \\
&= 0 - \nabla_{\mathbf{X}^{[n:n+n']'}}\hat{W}_\lambda(\mathbf{X}^{[0:n]}, \mathbf{X}^{[n':n+n']}), \\
&\nabla_{\mathbf{X}^{[n:n+n']'}}\hat{W}_\lambda(\mathbf{X}^{[0:n]}, \mathbf{X}^{[n':n+n']}) \quad \text{is not a function of } \mathbf{Y}
\end{aligned}
$$

Consequently, $\tilde{\mathbf{G}}' = \mathbf{G}^{[n:n+n']} \cdot \min\left(\frac{\Delta}{||\mathbf{G}^{[n:n+n']}||_2}, 1\right)$ does not contain information about $\mathbf{Y}$.

*(iii)* The post-processing property of differential privacy now guarantees that releasing $\hat{\mathbf{G}} = \text{concat}(\tilde{\mathbf{G}} + \gamma, \tilde{\mathbf{G}}')$ enjoys the same privacy protection as $\tilde{\mathbf{G}} + \gamma$. $\qquad\square$

## C Algorithms

---

**Algorithm 2** Poisson Sample

---

**Input** : $d = \{(\mathbf{y}, \mathrm{l}) \in \mathcal{X} \times \{0, ..., L\}\}^M$, sampling ratio $q$
**Output**: $\mathbf{Y} = \{(\mathbf{y}_j, \mathrm{l}_j) \in \mathcal{X} \times \{0, ..., L\}\}_{j=1}^m$, $m \geq 0$
$\mathrm{s} = \{\sigma_i\}_{i=1}^M \overset{i.i.d.}{\sim} \text{Bernoulli}(q)$
$\mathbf{Y} = \{d_j | \mathrm{s}_j = 1\}_{j=1}^m$

---

---

**Algorithm 3** Sinkhorn Algorithm $\hat{W}_\lambda(\mathbf{X}, \mathbf{Y})$

---

**Input:** $\mathbf{X} = \{\mathbf{x}\}^n, \mathbf{Y} = \{\mathbf{y}\}^m, \lambda$
**Output:** $W_\lambda$
$\forall (i, j), C_{[i,j]} = c(\mathbf{X}_i, \mathbf{Y}_j)$
$\mathbf{f}, \mathbf{g} \leftarrow \vec{0}$
$\hat{\mu}, \hat{\nu} \leftarrow \text{Unif}(n), \text{Unif}(m)$
**while** not converged **do**
$\quad \forall i, \mathbf{f}_i \leftarrow -\lambda \log \sum_{k=1}^m \exp(\log(\hat{\nu}_k) + \frac{1}{\lambda}\mathbf{g}_k - \frac{1}{\lambda}C_{[i,k]})$
$\quad \forall j, \mathbf{g}_j \leftarrow -\lambda \log \sum_{k=1}^n \exp(\log(\hat{\mu}_k) + \frac{1}{\lambda}\mathbf{f}_k - \frac{1}{\lambda}C_{[k,j]})$
**end while**
$W_\lambda = \langle \hat{\mu}, \mathbf{f} \rangle + \langle \hat{\nu}, \mathbf{g} \rangle$

---

## D Experiment Details

### D.1 Datasets

MNIST and Fashion-MNIST both consist of 28x28 grayscale images, partitioned into 60k training images and 10k test images. The 10 labels of the original classification task correspond to digit/object class. For calculating FID scores, we repeat the channel dimension 3 times. CelebA is composed of $\sim$200k colour images of celebrity faces tagged with 40 binary attributes. We downsample all images to 32x32, and use all 162,770 training images for training and all 19,962 test images for evaluation. Generation is conditioned on the gender attribute. We compute FID scores between our synthetically generated datasets of size 60k and the full test data (either 10k or 19,962 images). The MNIST dataset is made available under the terms of the Creative Commons Attribution-Share Alike 3.0 license. The Fashion-MNIST dataset is made available under the terms of the MIT license. MNIST and Fashion-MNIST do not contain personally identifiable information. The CelebA dataset is available for non-commercial research purposes only. It contains images of celebrity faces that are identifiable.

## D.2  Classifiers

For logistic regression, we use scikit-learn's implementation, using the L-BFGS solver and capping the maximum number of iterations at 5000. The MLP and CNN are implemented in PyTorch. The MLP has one hidden layer with 100 units and a ReLU activation. The CNN has two hidden layers with 32 and 64 filters, and uses ReLU activations. We train the CNN with dropout ($p = 0.5$) between all intermediate layers. Both the MLP and CNN are trained with Adam with default parameters while using 10% of the training data as hold-out for early stopping. Training stops after no improvement is seen in hold-out accuracy for 10 consecutive epochs.

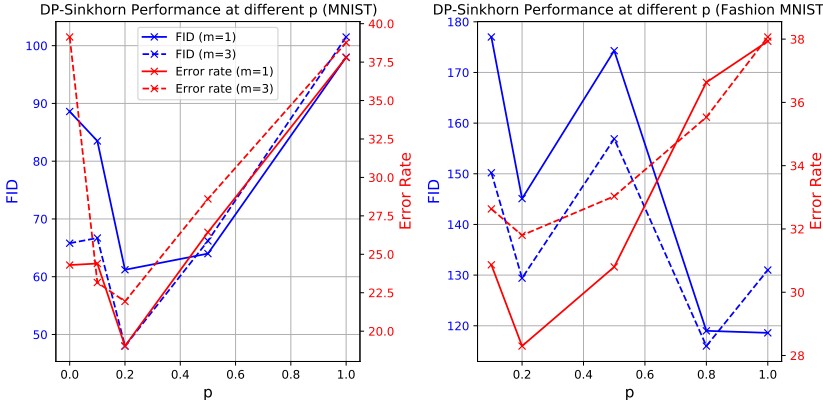

Figure 5: Effect of $p$ on DP-Sinkhorn performance. Left: performance on MNIST. Right: performance on Fashion MNIST. The performance is reported in terms of image quality (FID) and utility (error rate).

## D.3  Architecture, Hyperparameters, and Implementation

Our DCGAN-based architecture uses 4 transposed convolutional layers with ReLU activations at the hidden layers and tanh activation at the output layer. A latent dimension of 12 and class embedding dimension of 4 is used for MNIST and Fashion-MNIST experiments. CelebA experiments use a latent dimension of 32 and embedding dimension of 4. The latent and class embeddings are concatenated and then fed to the convolutional stack. The first transposed convolutional layer projects the input to $256 \times 7 \times 7$, with no padding. Layers 2, 3, and 4 have output depth $[128, 64, 1]$, kernel size $[4, 4, 3]$, stride $[2, 2, 1]$, and padding $[1, 1, 1]$.

Our BigGAN-based architecture uses 4 residual blocks of depth 256, and a latent dimension of 32. Each residual block consists of three convolutional layers with ReLU activations and spectral normalization between each layer. Please refer to [16] for more implementation details. Our implementation is based on https://github.com/ajbrock/BigGAN-PyTorch.

For semi-debiased Sinkhorn loss, we set $p = 0.2$ for results reported in Table 1 and Table 2. We used $m = 1$ and $m = 3$ for mixing $L_1$ with $L_2$ loss. Hyperparameter tuning results are reported in Table 4 and Table 5 and visualized in Figure 5. Setting $p = 0.2$ provides the best overall performance in terms of error rate and FID for both datasets.

To conditionally generate images given a target class $l$, we inject class information to both the generator and the Sinkhorn loss during training. For the loss function, we follow [42] and concatenate a scaled one-hot class encoding of class label $l$ to both the generated images and real images. Intuitively, this works by increasing the cost between image pairs of different classes, hence shifting the weight of the transport plan ($P_\lambda^*$ in Eq. 4) towards class-matched pairs. A scaling constant $\alpha_c$ determines the importance of class similarity relative to image similarity in determining the transport plan. Thus, $\mathbf{x}$ and $\mathbf{y}$ are replaced by $[\mathbf{x}, \alpha_c*\text{onehot}(l_x)]$ and $[\mathbf{y}, \alpha_c*\text{onehot}(l_y)]$ for class-conditional generation when calculating the element-wise cost.

Hyperparameters of the Sinkhorn loss used were: $\alpha_c = 15$, and entropy regularization $\lambda = 0.05$ in MNIST and Fashion-MNIST experiments. $\lambda = 5$ is used for CelebA experiments. We use

the implementation publically available at https://www.kernel-operations.io/geomloss/index.html and all other hyperparameters are kept at their default values. For all experiments, we use the Adam [50] optimizer with learning rate $10^{-5}$, $\beta = (0.9, 0.999)$, weight decay $2 \times 10^{-5}$.

### D.4 Implementation of Differential Privacy

For privacy accounting, we use the implementation of the RDP Accountant available in Tensorflow Privacy.[3] All experiments use Poisson sampling for drawing batches of real data, and are amenable to the analysis implemented in `compute_rdp`.

For MNIST and Fashion-MNIST results reported in the main body, we use a noise scale of $\sigma = 1.1$ and a batch size of 50 resulting in a sub-sampling ratio of $q = 1/1200$, which gives us $\sim 3.4$ million training iterations to reach $\varepsilon = 10$ for $\delta = 10^{-5}$. For the non-private runs, we use a batch size of 500, which improves image quality and diversity. When training with DP, increasing batch size significantly increases the privacy cost per iteration, resulting in poor image quality for fixed $\varepsilon = 10$. Image gradient perturbation used a clipping norm bound of 0.5, while the parameter gradient perturbation variant used a clipping norm bound of 1.

For CelebA results reported in the main body, we use a noise scale of $\sigma = 0.8$ and a batch size of 200 resulting in a sub-sampling ratio of $q = 0.00123$. At $\delta = 10^{-6}$, we train for 1.1 million steps to reach $\varepsilon = 10$.

**Computational Resources**  We perform experiments on an internal in-house GPU cluster, consisting of V100 NVIDIA GPUs. Each experiment is run on a single GPU with 16GB of VRAM. On MNIST and FashionMNIST, each epoch of training takes about 50 seconds to complete. Training of the generators to $(10, 10^{-5})$-DP takes 40 GPU hours to complete. On CelebA, experiments with the DCGAN architecture take $\sim$75 seconds per epoch during training, which totals to 12 GPU hours per run. BigGAN experiments take $\sim$250 seconds per epoch instead, totalling to $\sim$40 GPU hours per run.

We estimate the total amount of GPU hours used throughout this project to be $\approx$10,000 GPU hours. We assume that an average run takes around 40 GPU hours and each round of hyperparameter tuning experiments typically consists of 16 runs. From the conceptualization of the project to its current form, we performed 16 such parameter sweeps following changes to methodology, implementation, and parameter range. This totals to 10,240 GPU hours.

## E  Additional Results

We evaluate the impact of architecture choice on the performance in the CelebA task by comparing DP-Sinkhorn+BigGAN with DP-Sinkhorn+DCGAN, under $L_2$ loss. Results are summarized in Table 6 and visualized in Figure 6. Qualitatively, despite reaching lower FID score, the DCGAN-based generator's images have visible artifacts that are not present in models trained with BigGAN-generators.

Additional DP-Sinkhorn samples for MNIST and Fashion-MNIST are shown in Figures 7 and 8, respectively.

---

[3]https://github.com/tensorflow/privacy/

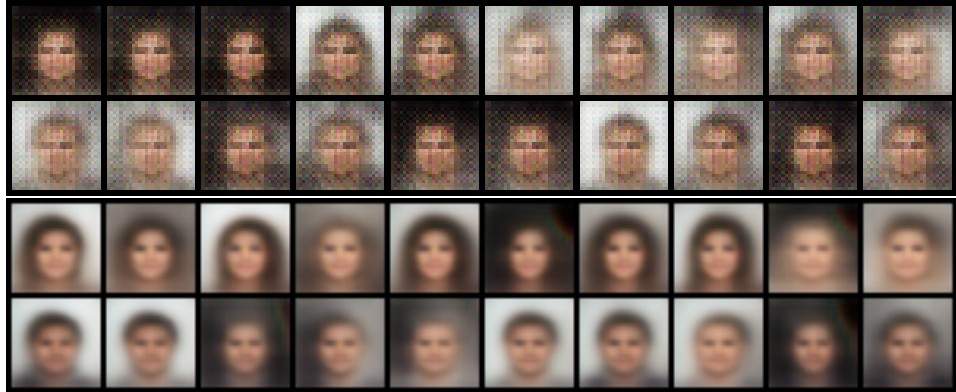

Figure 6: Additional DP-Sinkhorn generated images under $(10, 10^{-6})$differential privacy. Top two rows use DCGAN-based generator, while bottom two rows use BigGAN-based generator.

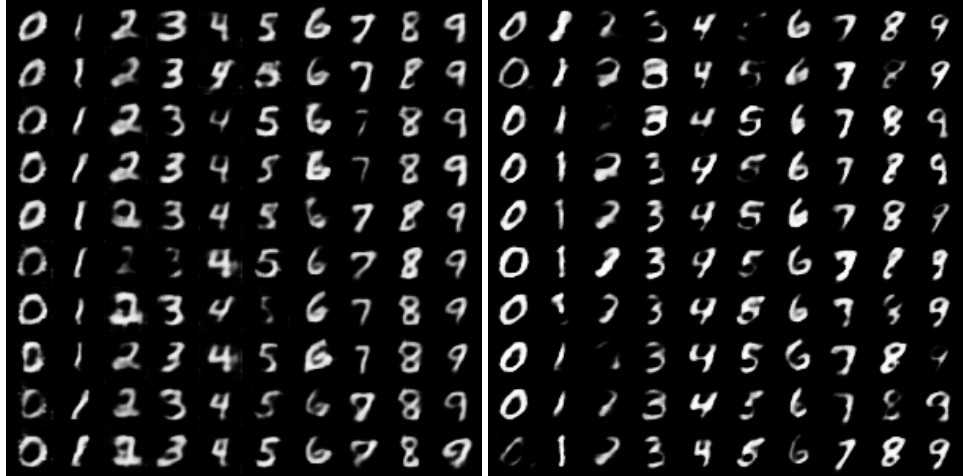

Figure 7: Additional images generated by DP-Sinkhorn, trained on MNIST. Left: Images generated by DP-Sinkhorn ($m = 1$); Right: Images generated by DP-Sinkhorn ($m = 3$).

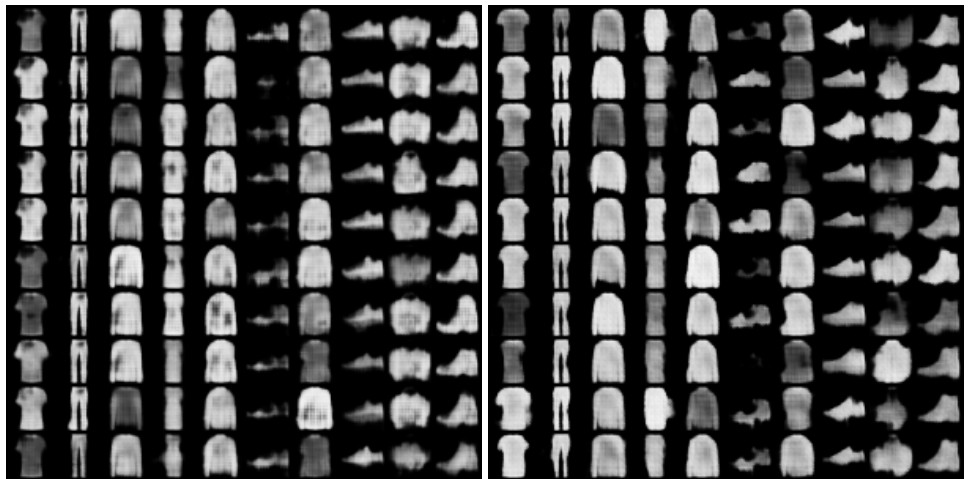

Figure 8: Additional images generated by DP-Sinkhorn, trained on Fashion-MNIST. Left: Images generated by DP-Sinkhorn ($m = 1$); Right: Images generated by DP-Sinkhorn ($m = 3$).

Table 4: DP-Sinkhorn ($\epsilon = 10$) hyperparameter search for $m$ and $p$ on MNIST

| $p$ | $m$ | FID | Acc (%) | | |
|-----|-----|-----|---------|---|---|
| | | | Log Reg | MLP | CNN |
| 1.0 | 0 | 103.1 | 60.6 | 62.4 | 64.0 |
| 1.0 | 0.3 | 106.8 | 67.6 | 65.2 | 66.1 |
| 1.0 | 1 | 98.0 | 64.7 | 63.0 | 60.5 |
| 1.0 | 3 | 101.5 | 61.3 | 61.4 | 61.0 |
| 0.5 | 0 | 62.8 | 75.2 | 76.0 | 80.2 |
| 0.5 | 0.3 | 65.9 | 72.1 | 72.4 | 75.7 |
| 0.5 | 1 | 64.0 | 71.8 | 72.1 | 76.8 |
| 0.5 | 3 | 66.2 | 71.4 | 70.0 | 72.8 |
| 0.2 | 0 | 101.8 | 80.3 | 81.0 | 78.3 |
| 0.2 | 0.3 | 91.3 | 79.6 | 80.0 | 80.3 |
| 0.2 | 1 | 61.2 | 79.5 | 80.2 | 83.2 |
| 0.2 | 3 | 48.0 | 78.2 | 76.3 | 79.6 |
| 0.1 | 0 | 141.0 | 78.2 | 80.9 | 79.3 |
| 0.1 | 0.3 | 89.3 | 77.9 | 78.6 | 74.4 |
| 0.1 | 1 | 83.5 | 75.1 | 77.4 | 74.3 |
| 0.1 | 3 | 66.7 | 76.1 | 76.7 | 77.7 |
| 0 | 0 | 124.3 | 79.3 | 82.0 | 80.8 |
| 0 | 1 | 88.6 | 74.4 | 76.6 | 76.1 |
| 0 | 3 | 65.8 | 62.1 | 56.4 | 64.1 |

Table 5: DP-Sinkhorn ($\epsilon = 10$) hyperparameter search for $m$ and $p$ on Fashion MNIST

| $p$ | $m$ | FID | Acc (%) | | |
|-----|-----|-----|---------|---|---|
| | | | Log Reg | MLP | CNN |
| 1.0 | 1 | 118.6 | 61.8 | 60.1 | 64.3 |
| 1.0 | 3 | 131.0 | 62.4 | 62.8 | 60.6 |
| 0.8 | 1 | 119.0 | 63.7 | 63.1 | 63.3 |
| 0.8 | 3 | 116.0 | 64.3 | 62.2 | 66.9 |
| 0.5 | 1 | 174.3 | 70.1 | 70.7 | 66.8 |
| 0.5 | 3 | 156.9 | 67.6 | 68.5 | 64.8 |
| 0.2 | 0.3 | 170.9 | 69.1 | 69.2 | 66.5 |
| 0.2 | 1 | 145.1 | 73.0 | 72.8 | 69.3 |
| 0.2 | 3 | 129.4 | 70.2 | 70.2 | 64.2 |
| 0.1 | 0.3 | 170.9 | 69.1 | 69.2 | 66.5 |
| 0.1 | 1 | 177.0 | 69.8 | 70.3 | 67.3 |
| 0.1 | 3 | 150.2 | 67.9 | 68.4 | 65.8 |

Table 6: Differetially private image generation results on downsampled CelebA.

| Method | DP-$\epsilon$ | FID | Acc (%) | |
|--------|---------------|-----|---------|---|
| | | | MLP | CNN |
| Real data | $\infty$ | 1.1 | 91.9 | 95.0 |
| DCGAN+DP-Sinkhorn | 10 | 156.7 | 74.96 | 74.62 |
| BigGAN+DP-Sinkhorn | 10 | 168.4 | 76.18 | 75.79 |