# OpenReview forum: "Don’t Generate Me: Training Differentially Private Generative Models with Sinkhorn Divergence"
_NeurIPS.cc/2021/Conference — NeurIPS 2021 Poster_

### Official Review · Reviewer_BgbF · 2021-07-13

**Rating:** 5
**Confidence:** 3

**Summary:**

The paper studies the problem of training generative models under differential privacy constraint. To this end, they propose a new optimal transport based objective, the semi-debiased Sinkhorn loss, analyze its privacy properties, and evaluate its performance on a number of image datasets.


**Limitations And Societal Impact:**

Yes

**Main Review:**

Originality: The choice of optimizing the semi-debiased Sinkhorn loss is, to the best of my knowledge, novel. The task itself (differentially private generative models) and techniques (gradient clipping +  adding Gaussian noise) are well-known in the literature. While the authors do an adequate job of surveying related work, their exposition as to other methods could be significantly improved in the experiments section (Table 1). It’s not clear exactly how the objectives and training procedures differ between these methods and those proposed by the authors.  A quick summary of how these other algorithms work would be beneficial so as to better understand the contributions of their paper.

Quality: The submission is technically sound. The proofs appear to be correct, although I was not able to verify them in precise detail. The methods used in evaluation seem appropriate, although it would be helpful to the reader if the author’s provided confidence intervals for the results in table 1 to get a better sense of the variance in these methods.

Clarity: The paper is for the most part well written.

While their methods do appear to provide some improvement on MNIST and CelebA, these datasets are relatively low dimensional and it would be valuable to understand how the procedures introduced in this paper perform in more complicated settings. Given that the theoretical contributions are based on standard applications of DP tools, the overall impact of the work seems to me like it hinges on the quality of the empirical results. To this end, I would have liked to see experiments in a wider variety of domains to be better convinced of the impact that these ideas will have in practice.





**Time Spent Reviewing:**

5

---

> ### Author Response · Authors · 2021-08-10
> **Author response to reviewer BgbF**
>
> We thank the reviewer for the detailed feedback.
>
> **Regarding comparison to other methods**, our method stands out from existing methods in that we train a generator with a primal form optimal transport (OT) divergence, which does not depend on adversarial training schemes. We provide a more detailed overview of existing methods here.
>
> DPGAN [1] first introduced the idea of combining differential privacy with GANs in a simple scheme where DPSGD is used when updating the generator. This is followed up a year later by dpGAN [2], which adds a decaying clipping threshold that heuristically matches the decreasing gradient magnitude during training. DP-CGAN [7] adds class conditioning to DPGAN for generation of conditional data. PATE-GAN [3] adopts the PATE framework for generative learning by applying the PATE framework to train a private student discriminator; only generated images are scored by this student discriminator to train the generator. This work is improved by G-PATE[4], which uses random projections and gradient quantization to directly aggregate discriminator gradients for updating the generator. Importantly, G-PATE makes the point that only the generator is released in the differentially private (DP) generative learning task, thus a large (~1000s) ensemble of non-private discriminators can be used to train a private generator. GS-WGAN[5] brings this idea back to DPSGD-based training, in which the gradient aggregate from an ensemble of discriminators is processed by the Gaussian mechanism. Unlike DP-GAN, this is performed on the image gradient, which has fewer dimensions than the parameter gradient. Datalens[6] further improves upon G-PATE by introducing TopAgg - a three step gradient compression and aggregation algorithm, which provides stable, quantized discriminator gradients at a low privacy cost. Lastly, DP-MERF[8] uses random fourier features to construct class prototypes for each class, and then train a generator to match the class prototypes in the random fourier feature space. We will expand the discussion of related works in the final version of the paper.
>
> Our method, DP-Sinkhorn, inherits techniques such as image gradient perturbation from this existing literature, but takes a markedly distinct approach through questioning the necessity of complex adversarial learning schemes in DP generative modeling. We use as loss function the Sinkhorn-divergence, a primal-form approximation to the OT distance, which faithfully approximates the true Wasserstein distance. A direct benefit of this change is that our method does not require training thousands of discriminators. We also introduce the semi-debiased Sinkhorn loss to address the bias-variance tradeoff that arises during training. Overall, we present a method that is methodologically distinct in the current landscape of DP generative learning, and exceeds state of art performance in this task. In our experiments, we compare DP-Sinkhorn to the more recent methods ([4,5,6,7,8]) outlined above.
>
> **Regarding the choice of datasets**, we evaluate DP-Sinkhorn on MNIST and FashionMNIST as these are the most widely used benchmarks for the task of DP generative modeling. DP-Sinkhorn exceeds state-of-the-art performance on both datasets. In addition, we include experiments on CelebA, which demonstrate the extensibility of DP-Sinkhorn to RGB datasets. On CelebA, DP-Sinkhorn generates RGB image data that reasonably resembles the original data, whereas existing methods either do not work on RGB images at all (GS-WGAN collapses) or produce indiscernible images (Datalens). We believe that DP-Sinkhorn already represents the best performance achieved by DP generative models in complex RGB images. Extending DP-Sinkhorn to even larger scale datasets is an exciting direction for future work.
>
> We will extend the related works section of the paper with the discussed material. __We hope this reply addresses all concerns raised in the review. If so, we kindly ask the reviewer to raise their review score.__ We would be happy to discuss any other concerns with the reviewer.
>
>
> [1] L. Xie, K. Lin, S. Wang, F. Wang, and J. Zhou, “Differentially private generative adversarial network,” arXiv preprint arXiv:1802.06739, 2018.
>
> [2]L. Frigerio, A. S. de Oliveira, L. Gomez, and P. Duverger, “Differentially private generative adversarial networks for time series, continuous, and discrete open data,” in IFIP International Conference on ICT Systems Security and Privacy Protection, pp. 151–164, Springer, 2019
>
> [3] Jordon J, Yoon J, Van Der Schaar M. “PATE-GAN: Generating synthetic data with differential privacy guarantees.” In International conference on learning representations 2018 Sep 27.
>
> [4]Y. Long, S. Lin, Z. Yang, C. A. Gunter, H. Liu, and B. Li, “Scalable differentially private data generation via private aggregation of teacher ensembles,” 2019.
>
> [5] D. Chen, T. Orekondy, and M. Fritz, “GS-WGAN: A Gradient-Sanitized Approach for Learning Differentially Private Generators,” in Advances in Neural Information Processing Systems, 2020
>
> [6] B. Wang, F. Wu, Y. Long, L. Rimanic, C. Zhang, and B. Li, “Datalens: Scalable privacy preserving training via gradient compression and aggregation,” arXiv preprint arXiv:2103.11109, 2021
>
> [7]  R. Torkzadehmahani, P. Kairouz, and B. Paten, “Dp-cgan: Differentially private synthetic data and label generation,” in Proceedings of the IEEE Conference on Computer Vision and Pattern Recognition Workshops, pp. 0–0, 2019.
>
> [8] F. Harder, K. Adamczewski, and M. Park, “Differentially private mean embeddings with random features (dp-merf) for simple & practical synthetic data generation,” arXiv preprint arXiv:2002.11603, 2020.

---

### Official Review · Reviewer_oTPG · 2021-07-14

**Rating:** 7
**Confidence:** 3

**Summary:**

The paper considers the problem of generating private synthetic data using generative adversarial networks trained with differential privacy. While this has been considered, the authors propose an improvement by optimizing Sinkhorn divergence, which has been shown to be a successful strategy for nonprivate GANs. Due to challenges which do not arise in nonprivate GAN training (a difficult bias variance tradeoff), they use a weakening of the unbiased Sinkhorn divergence. They evaluate their strategy on multiple datasets, measuring both FID and a classifier's performance on synthetic data.

**Limitations And Societal Impact:**

Limitations and societal impact are fine.

**Main Review:**

Pros:
The experimental results are strong (improving over existing DP strategies for synthetic data), although evaluated on small datasets. I'm a bit surprised that the classifier performance on MNIST is so bad, when even at somewhat small epsilon values, just because of how easy MNIST is.

Being able to avoid noising G' is nice.

The authors put a solid amount of effort into ablation to justify their strategy.

Cons:
Epsilon is really large - the typical way of justifying this is showing that privacy attacks are not effective - could you do this here? I have seen, for example, membership inference being run on discriminators to test the privacy of generative models.

The l_1 and l_2 losses are pretty simple for image data; I have seen perceptual distance used in place of these for image generation.

I'd be interested in seeing results on non-ML images, e.g. medical image datasets or medical records. This seems to be fairly standard in the DP synthetic data literature.

Overall, I am happy with the paper. I'm not super familiar with the generative model literature, but it seems to take a known nonprivate technique and apply it, overcoming challenges that pop up.

**Time Spent Reviewing:**

3

---

> ### Author Response · Authors · 2021-08-10
> **Author response to reviewer oTPG**
>
> We thank the reviewer for the detailed comments. We respond to specific points in the review below:
> 1. **Regarding choice of $\epsilon$ value**, we use $\epsilon=10$ for our benchmarks to enable direct performance comparisons with existing methods. At $\epsilon=2.3$ (Fig. 4b), the improvement of DP-Sinkhorn over GS-WGAN, one of the strongest baselines, is even greater than that at $\epsilon=10$. We appreciate the reviewer’s suggestion about membership inference attacks. An advantage of DP-Sinkhorn over GANs is the absence of a discriminator. As such, it is not vulnerable to existing white-box attacks that target the discriminator’s tendency to overfit the training set. On the other hand, black-box attacks (incl. Monte Carlo-based multi-sample attacks) have found limited success against GANs [1,2]. It is thus possible that the success rate of membership inference attacks does not change significantly between DP and non-DP generators. We think this is an interesting direction for future work.
>
> 2. **Regarding perceptual distances**, we have indeed considered using more sophisticated distance metrics. One option for perceptual distance would be to use a pre-trained feature extractor to extract features of real and generated images. However, this pre-training must be performed on public data to maintain privacy protection, which complicates the comparison with existing methods. We have also experimented with an adversarial feature extractor that is trained on the private dataset. We found that adding this feature extractor improves performance in non-private settings, but degrades performance in the DP setting due to added complexity and higher privacy cost.
>
> 3. We appreciate the suggestion to experiment on medical datasets. We are not aware of any DP generative method that has been applied to medical images, and think it would be an important direction for future work. For medical records datasets, combining shallow models with existing methods already produces good results with small privacy budgets (e.g. [3]), while our focus is on the image domain, which is more difficult due to its high dimensionality and strong correlation between dimensions (pixels).
>
> We would be happy to answer any further questions during the discussion period.
>
> [1]Hayes J, Melis L, Danezis G, De Cristofaro E. Logan: Membership inference attacks against generative models. arXiv preprint arXiv:1705.07663. 2017 May 22.
>
> [2]Hilprecht, B., Harterich, M., and Bernau, D. Monte carlo and reconstruction membership inference attacks against generative models. Proceedings on Privacy Enhancing Technologies, 2019(4):232–249, 2019
>
> [3] Chin-Cheong, Kieran & Sutter, Thomas & Vogt, Julia. (2020). Generation of Differentially Private Heterogeneous Electronic Health Records.

---

### Official Review · Reviewer_Xbc7 · 2021-07-15

**Rating:** 6
**Confidence:** 3

**Summary:**

This paper proposes a new technique for generating differentially-private synthetic data. The approach avoids adversarial training altogether, which has historically given poor model accuracy due to its instability and sensitivity to noise. Instead, the authors minimize Sinkhorn divergence, which is a computationally-tractable approximation to the optimal transport distance between the true distribution and the generated one. The authors demonstrate empirical gains over existing techniques for DP generative models.

**Limitations And Societal Impact:**

The authors seem to be aware of several of the limitations of their approach. It may be helpful to discuss the fact that the non-private model has very poor fidelity, but the DP version still outperforms existing baselines. Is this fundamental? I think this is an interesting question to explore.

**Main Review:**

Overall, I think this is a potentially interesting idea, and I believe there is a lot of value to pursuing non-adversarial techniques for DP generative models. The idea of using sinkhorn divergence to (non-adversarially) train a generative model appears to be new, to the best of my knowledge.

My main concern is that the proposed generative model simply doesn’t work very well in the non-private setting, as evidenced by its poor FID scores. This seems to suggest that there is no hope of making this approach ever have “good enough” fidelity to compete with non-DP data. That being said, Sinkhorn-DP does seem to have slightly better FID and prediction accuracy than the many competing baselines, so that is an interesting observation. Though the difference is not large compared to, say, GS-WGAN. So when we’re talking about FID scores that in the hundreds, I would have ideally liked to see a more substantial improvement over the state-of-the-art scores.

Technically, I found the ideas in this paper to be a bit limited. The approach is heuristic, and there is no fundamental reason this approach should work better, other than getting rid of adversarial training. The Sinkhorn-DP approach seems to be very sensitive to hyperparameters as shown in the evaluation, and it is only evaluated on small-scale datasets. The theoretical claims are limited to proving DP of the scheme, which follows trivially from the construction. So in terms of contribution, I am not sure this paper deepens our understanding of how to improve DP generative models. That being said, it does appear to give (modest) gains over SOTA.

I was not able to understand a few details of the approach from the paper:

Def. 4.1 – how is the cost matrix C_AB defined, and what is its intuitive meaning? (I did see you partially discuss in this in Sec. 4, but it should come earlier, and I still wanted more details on how to interpret C_AB). What is the Sinkhorn Algorithm used to compute P_AB, and why isn’t W_λ(X, X)=0?

Algorithm 1: Why do you clip grad(X[n:n+n’]) if you are not adding DP noise? What is the meaning of the categories 0,…, L? Are you assuming labelled data and conditional generation? If so, this isn’t described in the problem setup… (Again, I later saw in the evaluation that you are considering conditional generation, but this wasn’t clear upfront)

Minor comments:
In contrast, only a single generator network is trained in DP-Sinkhorn, making our approach more amenable to various hardware configurations -> do you mean a single discriminator? The previous lines are about multiple discriminators.


**Time Spent Reviewing:**

2.5

---

> ### Author Response · Authors · 2021-08-10
> **Author response to reviewer Xbc7**
>
> We thank the reviewer for the detailed feedback. We address the main points of the review below:
>
> **On performance of DP generative models**: We think comparing differentially private (DP) generative models by their non-DP performance is not very conducive to developing better DP generative models due to differences in these domains. Training a DP generative model requires a large amount of noise to be added during optimization, which often necessitates reducing model capacity in favor of simpler models that are easy to train. For instance, Datalens [1] aggressively compresses discriminator gradients into binary vectors in order to reduce information leakage; GS-WGAN [2] uses an ensemble of weak discriminators, where each discriminator is only trained on a small split of data. In both cases, these techniques are certainly detrimental to the performance in the non-private setting, but benefit their respective methods in the privacy-preserving setting. In our preliminary experiments, we experimented with using adversarially trained feature extractors for computing the cost function, yet found their performance under the DP setting to be worse than simple L2-based cost functions. These examples show that non-DP performance does not necessarily correlate with DP performance. Generally speaking, the rigorous privacy constraints by DP make strict DP generative modeling significantly harder than regular generative modeling. Training behaves quite differently and requires new ideas and tailored methods.
>
> As an aside, using a similar generator architecture as ours, [2] obtains accuracies of 84%/77% on MNIST/FashionMNIST with a non-DP GAN and MLP classifier, which is similar to the MLP accuracy (89%/79%) of our non private Sinkhorn-based generative model. This shows that learning with Sinkhorn divergence is not inherently worse than adversarial training. Apart from performance advantages in the DP domain, DP-Sinkhorn is also more robust to learning rate choices, requires less VRAM to train, and offers stronger low-epsilon (stricter privacy) performance.
>
> **Regarding novelty of approach**: The method that we propose in the paper goes beyond the simple composition of DP and Sinkhorn generative learning. We emphasize that DP-Sinkhorn not only outperforms existing DP generative methods, but also baselines that use fully-debiased or non-debiased Sinkhorn losses (corresponding to $p=1$ and $p=0$ respectively). We further study the effect of semi-debiasing experimentally in Section 5.2 and Figure 4(c) and find that adjusting the debiasing parameter $p$ can help achieve a balance between variance and bias, which is particularly important for the DP setting as argued in Section 4.2. Furthermore, DP-Sinkhorn can generate RGB image data that reasonably resembles the original data (on CelebA), whereas existing methods either do not work on RGB images (GS-WGAN) or produce indiscernible images (Datalens).
> Our experiments (Figures 4a and 4b) find DP-Sinkhorn to be *less* sensitive to hyperparameter choices than GS-WGAN, demonstrating the advantages of non-adversarial training.
>
> **Regarding definitions in 4.1**:
> 1. Thank you for pointing this out! For two sets of examples $A={x_i}^n$ and $B={y_j}^m$, the cost matrix $C_{AB} \in R_{+}^{n \times m}$ is defined as $C_{i,j} = c(x_i, y_j)$, where $c$ is the cost function (in our case, squared $L2$ distance). We will add this definition into the manuscript. Intuitively, it represents how similar each example in $A$ is to each example in $B$ as measured by $c$.
> 2. The Sinkhorn algorithm optimizes the dual potential functions $f$ and $g$ (Equation 10 in supplementary materials) to find the optimal transport plan. Intuitively, the Sinkhorn algorithm iteratively computes $f$ and $g$ through the relation $f_i = -\lambda \log \sum_{j=1}^n (\log \beta_j + \frac{1}{\lambda} (g_j - C_{i,j}))$ and $g_j = -\lambda \log \sum_{i=1}^n (\log \alpha_i + \frac{1}{\lambda} (f_i - C_{i,j}))$. This algorithm is thoroughly explored in chapter 4.1 of [3].
> 3. The second term in Equation 1 is the entropy regularization on the transport plan $\pi$. Specifically, it is $\lambda$ times the KL divergence between $\pi$ and $\alpha \times \beta$, which can only be zero when $\pi = \alpha \times \beta$, meaning the optimal transport plan is uniform over all elements. This only occurs when all elements of X are identical to each other such that elements of the cost matrix are uniform.
>
> **Regarding Clipping** We clip the “self” group gradients $G^{[n:n+n’]}$ such that its magnitude does not dominate the concatenated gradient vector which also contains the “cross” group gradients that were clipped and noised for privacy purposes. Our method can be used for either conditional or unconditional generation. We choose to emphasize this in the experiments section as it is important for purposes of benchmarking. We will make it more clear in the manuscript that we are considering the task of conditional generation.
>
> **Regarding minor comment**: Here we are making the comparison that GS-WGAN uses 1 generator and K ($\sim10^3$) discriminators, while DP-Sinkhorn only requires a single generator and no discriminator at all (the objective is calculated via the Sinkhorn divergence).
>
> We will add additional details to the manuscript to make sure the above points are clearly explained in the final version of the paper.
>
> __If our response addresses the concerns of the reviewer, please consider raising the review score.__ We would be glad to discuss any additional questions during the discussion period.
>
> [1] B. Wang, F. Wu, Y. Long, L. Rimanic, C. Zhang, and B. Li, “Datalens:  Scalable privacy pre-serving training via gradient compression and aggregation,”arXiv preprint arXiv:2103.11109,2021.
>
> [2] D. Chen, T. Orekondy, and M. Fritz, “GS-WGAN: A Gradient-Sanitized Approach for Learning Differentially Private Generators,” in Advances in Neural Information Processing Systems, 2020
>
> [3] G. Peyre and M. Cuturi, “Computational Optimal Transport,” Foundations and Trends in Machine Learning, vol. 11, no. 5-6, pp. 355–607, 2019.

---

> > ### Comment · Reviewer_Xbc7 · 2021-09-11
> > **Response**
> >
> > Thank you to the authors for the clarifications and discussion. After some more thought and discussion, I agree with your point that performance in the non-private setting is not necessary for DP performance. I guess my main concern is that the non-private performance seems to be so poor that there isn't enough room for improvement to any reasonable level. This is a little concerning, and it suggests to me that this approach is unlikely to be the right approach for this problem. However, perhaps it will spur other ideas that can achieve better performance. For this reason, I am raising my review score to a 6.

---

### Official Review · Reviewer_GTXj · 2021-07-15

**Rating:** 4
**Confidence:** 4

**Summary:**

The paper proposes DP-Sinkhorn, an optimal-transport based generative model for privacy-preserving data generation based on differential privacy.

Optimal transport based generative models minimize variants of the Wasserstein distance, in particular an entropy-regularized version. The Sinkhorn divergence adds an autocorrelation term which cancels out the entropic bias completely when the distance is zero. [36] showed that the gradients of the empirical Sinkhorn loss are biased and proposes using independently drawn samples from the generator, the "debiased" Sinkhorn loss.

This paper proposes the "semi-debiased" Sinkhorn loss which, instead of using independently drawn samples, splits the sample along the batch dimension.

Differential privacy is added by perturbing gradients with the privacy level determined using the RDP accountant.

The method of using the semi-biased Sinkhorn loss with DP is termed DP-Sinkhorn and the paper argues that the lack of adversarial training, as opposed to GANs, avoids training instability and the need for early stopping.

Experimental results indicate better performance on MNIST, Fashion-MNIST and Celeb-A compared to other DP generative models.

**Limitations And Societal Impact:**

There is not much discussion of limitations

**Main Review:**

In general, I found the paper to be confusing in terms of what it is arguing its main contributions are.

The paper casts itself as a contribution in privacy-preserving generative models, but there is not really any novelty in the DP domain as existing privacy mechanisms are used to achieve DP in the same way they are used for existing DP GANs and related models.

Rather, the primary novelty is that the authors have proposed the "semi-debiased" Sinkhorn loss, compared to the debiased Sinkhorn loss, but it's not clear what this has to do with privacy - and if this does represent a significant improvement, the paper does not dedicate much theory or experiments in attempting to understand its improvement (why the semi-debiased version is better in a general, non-private context).

This same confusion extends to the experimental results. While the results appear strong, the only included baselines are other DP generative models so it's unclear why the performance is better (is the underlying model just better in a non-private context so it remains better even after DP is added?). This confusion makes it difficult to judge the novelty and significance.

It seems there are 3 possibilities: 1) the paper is arguing OT-based generative models are better in general than GANs so they remain better when coupled with standard DP mechanisms; 2) the paper is arguing that the "semi-debiased" Sinkhorn loss significantly improves over the debiased Sinkhorn loss and that's what's responsible for the gains (making privacy a secondary thought); 3) the paper is arguing there is something specific about the semd-debiased Sinkhorn related to privacy that causes it to perform better (compared to other Sinkhorn losses) when coupled with DP mechanisms. I'm not sure which the paper is intending to argue and what the primary support would be for the claim.

I also found the claim that DP-Sinkhorn "avoids the need for early stopping" strange since this does not appear to be something specific to adversarial learning.

Finally, the writing in section 4.2 should be more clear and explicit since it's describing the main contribution, e.g. X' is never defined. It still remains unclear to me why the performance should be that different by simply splitting the batch rather than sample from the generator twice.

**Time Spent Reviewing:**

3-4

---

> ### Author Response · Authors · 2021-08-10
> **Author response to reviewer GTXj**
>
> We thank the reviewer for their detailed feedback.
>
> **Regarding the main contribution of the paper**, we are the first to adapt generative learning with Sinkhorn divergence to the task of differentially private (DP) image generation, and show that our particular design (semi-debiased DP-Sinkhorn) exceeds state-of-the-art performance. This is indeed similar to option 3) outlined by the reviewer, but we note that the performance of our method is superior to not only other methods designed for DP image generation, but also to alternative implementations of “Sinkhorn” + “DP”, as evidenced by our ablations. Figure 5 in the supplementary material compares the performances when using various debiasing values $p$, while Tables 2, 4 and 5 tabulate the performance under various hyperparameter settings. Setting $p=1$ produces an unbiased estimator, while setting $p=0$ gives a non-debiased loss estimator. It turns out our novel semi-debiasing is crucial in the DP setting to achieve strong performance. We believe that the current experiments provide strong empirical evidence for DP-Sinkhorn being the best performing one among current DP-generation methods.
>
> **To answer how semi-debiasing helps DP-Sinkhorn achieve SOTA performance**, we first wish to clarify to the reviewer that semi-debiasing is not just “splitting the sample along the batch dimension”. Instead, for batch size $n$, we generate $n*(1+p)$ samples, which are split in the batch dimension into two groups of size $n$, overlapping in the middle. This way, setting $p=1$ is equivalent to using two independently drawn batches, while setting $p=0$ means using the same batch twice.
>
> We motivate this semi-debiased loss by the need to balance bias-variance trade-off in the manuscript, and validate this claim experimentally. Figure 4c elucidates the bias-variance trade-off of the gradient estimator
> $G_{p} = \nabla_\theta \hat{S} (X(\theta),Y)$
> under different values of $p$. In more detail, for each value of $p \in \{0, 0.2, 0.4, 0.6, 0.8, 1\}$, we compute $G_{p}$ on three hundred batches of real and generated data to obtain its average and sample variance. As $G_{1}$ is unbiased, we use it as the ground truth. We plot the bias and variance against $p$ along the x-axis. We observe two prominent trends from this graph. First, as we increase $p$, bias decreases and variance increases. This requires us to find a balance in the trade-off between bias and variance. Second, we see flatter curves for generators trained with smaller $p$. As $p$ affects bias and variance through changing the number of resampled generated images, we can deduce that training with smaller $p$ likely results in greater similarity between generated images, which improves consistency across generated images at the cost of diversity. That is, if the generator is mode collapsed, $p$ would have no effect on the bias-variance trade-off, as resampling the latent variables would produce the same images. However, we note that previous works in the non-private domain have found fully debiased ($p=1$) Sinkhorn loss to provide high performance [1], whereas our experiments in the privacy preserving domain find a small amount of debiasing ($p=0.2$) to perform best. We think the main reason for this difference is that training in the privacy preserving domain is restricted in batch size and number of iterations, making the increased variance of the fully debiased loss more detrimental. As such, we introduced semi-debiasing in section 4.2 as a specific solution to the privacy preserving setting. Generally, it is worth keeping in mind that the rigorous privacy constraints by DP make strict DP generative modeling significantly harder than regular generative modeling. Training behaves quite differently and requires new ideas and tailored methods. We will try to better explain these aspects in the final version of the paper.
>
> **With respect to early stopping**, we are referring to the fact that GANs (incl. WGAN-GP) often exhibit non-convergence and instability [2], and early stopping before catastrophic divergence is often required in practice to obtain good generators (e.g. [3]). Training with Sinkhorn divergence does not exhibit the same kind of instability as GANs, therefore we ‘avoid the need for early stopping’. We acknowledge that this wording was maybe not ideal and we will improve this explanation in the final paper version.
>
> **Regarding notations**, we will add the definition of $X’$ to the manuscript. The key difference between Equation 5 (fully debiased empirical Sinkhorn) and Equation 6 (semi debiased empirical Sinkhorn) is that only $n’$ examples are resampled in 6, while $n$ examples are resampled in 5. We are effectively interpolating between sampling the generator twice ($p=1$) and using the same batch ($p=0$). This results in the bias-variance tradeoff as discussed above.
>
> __If our response answers the reviewer’s questions about the paper, we kindly ask the reviewer to reconsider their paper score.__ We would also be very happy to answer any further questions and concerns the reviewer may have during the discussion period.
>
> [1] T. Salimans, H. Zhang, A. Radford, and D. Metaxas, “Improving GANs using optimal transport,” in International Conference on Learning Representations, 2018.
>
> [2]Mescheder, Lars M., Andreas Geiger and S. Nowozin. “Which Training Methods for GANs do actually Converge?” ICML (2018).
>
> [3 ]Brock A, Donahue J, Simonyan K. Large scale GAN training for high fidelity natural image synthesis. arXiv preprint arXiv:1809.11096. 2018 Sep 28.

---

### Official Review · Reviewer_U7YX · 2021-07-16

**Rating:** 5
**Confidence:** 3

**Summary:**

The paper presents a DP model based on Sinkhorn GAN. Different from DP GAN models that are based on adversarial loss, they use an optimal transport (OT) distance similar to Wasserstein distance called Sinkhorn divergence. The idea is simple and incremental, it is an extension of Sinkhorn GAN. Experimental results looks promising but some important results are missing.


**Ethics Review Area:**

["I don’t know"]

**Limitations And Societal Impact:**

It is better if the authors mention few limitations and future directions for this model

**Main Review:**

- In Section 2, the authors briefly mentioned some of the existing DP-GAN methods based on DP-SGD or PATE. Many these techniques or new developed version of them are already applied noised on the gradient of the Wasserstein distance and handled instability of traditional GANs. It is not clear how the proposed method is different from the state-of-the-art techniques. Maybe a good comparison between DP-Sinkhorn and DP-WGAN can better address this issue.

- Please explain how DP-Sinkhorn can be applied on non-image data (e.g., tabular data) where both continues and categorical features exist. For GANs, there are several techniques that handle tabular data (e.g., CTGAN).

- Please provide more information about cross group and debiasing group and why noise is only added to the cross group?

- One of the disadvantage of GAN models are the long training time. It would be great if the authors show the training time comparison between the proposed model and DP-GAN models.

- In experimental results, why CelebA data is missing in table 1 (comparison results)? Why for this dataset, the image size is reduced to 32*32? How the model can handle large scale datasets (images with high resolution)? Why Figure 3 does not include other benchmarks?

- Please explain how the proposed model performs on imbalanced datasets and if it suffers from mode collapse similar to some traditional GAN models?

**Time Spent Reviewing:**

3

---

> ### Author Response · Authors · 2021-08-10
> **Author Response to Reviewer U7YX:**
>
> We thank the reviewer for the detailed comments. We address each point in the main review below:
>
> 1. **On differences between Sinkhorn and Wasserstein GAN** We would like to emphasize that the Sinkhorn divergence approximates the primal form of the Wasserstein distance, while Wasserstein GAN approximates the dual form. We have briefly discussed the differences between these two approaches in Section 4.2. Advantages of the primal form approximation include numerical stability, a more faithful approximation of the Wasserstein distance, and extensibility to higher-order Wasserstein distances. A more focused discussion on behavioral differences between primal vs dual can be found in [1].
> Computationally, the Sinkhorn divergence is computed as three matrix products between the optimal transport plan and the cost matrix (Eqs. 5 and 6), whereas Wasserstein GAN’s loss is the average output of its discriminator which approximates the dual potential function.
>
>     DP-Sinkhorn extends upon standard Sinkhorn divergence with debiasing and gradient perturbation for private generation of image data. Experimentally, we dedicate Figures 4a and 4b to compare the hyperparameter sensitivity between DP-Sinkhorn and GS-WGAN (the best performing implementation of DP-WGAN that we are aware of). We find that not only does DP-Sinkhorn outperform GS-WGAN in their best configurations (as reported in Table 1, and also in Figure 4b for various privacy parameters $\epsilon$), DP-Sinkhorn is also more robust to different hyperparameter choices. We have also expanded our related works section in the manuscript with descriptions of the existing methods. Note that to the best of our knowledge, we are the first to explore differentially private generative modeling using the primal form of the Wasserstein distance.
>
> 2. **Regarding tabular data**: We thank the reviewer for pointing out this interesting direction. DP-Sinkhorn can indeed be applied to tabular data in a fashion similar to CTGAN[2]. The generator architecture of CTGAN can be used to replace the CNN generator to produce categorical and floating-point data. Since our main focus in this paper is to present DP-Sinkhorn as a method for training generative models, not specific to any generator architecture, we opt to use a similar architecture as GS-WGAN. We perform experiments on MNIST and FashionMNIST, which are the most widely used benchmarks when developing novel differentially private generative models. However, note that DP-Sinkhorn may be even more easily applied to non-image data than other differentially private generative models, since it does not require a discriminator tailored to the data type. As our focus in the paper is on generating high-dimensional image data, we leave this exploration for future work.
>
> 3. **The cross and debiasing groups** refer to $\mathbf{X}^{[0,n]}$ and $\mathbf{X}^{[n,n+n’]}$, respectively, as used in Equation 6. The gradient with respect to the cross group contains information about the sensitive real data, while gradient with respect to the debiasing group does not. Hence, gradient perturbation is only required for the cross group. We prove the privacy guarantee of this method in Appendix B.1.
> 4. **In terms of training time**, while DP-Sinkhorn requires less parameters than comparable DP-GANs due to the absence of a discriminator, we cannot fairly say whether our method is faster to train than others due to the use of gradient perturbation with privacy accounting. The number of iterations is primarily determined via the privacy constraints and depends on the overall privacy budget, training batch size, and choice of noise scale $\sigma$ (note that this is a unique feature of training differentially private generative models, where each training iteration eats into our privacy budget). Training time can be made arbitrarily long or short by adjusting these parameters. This is true for both our method and other DP generative methods. We will amend the claims regarding training time in the manuscript. However, we do note that our method requires less memory to train than GS-WGAN, which uses an ensemble of discriminators and therefore has limited scalability to very large data sets.
>
> 5. **Regarding higher dimensional datasets** We did not replicate baseline methods for training on CelebA as their performance on the MNIST dataset was already poor. We attempted to train GS-WGAN on CelebA but could not obtain meaningful results as generators collapsed quickly to all white or black pixels. As such, we opted not to place these GS-WGAN images in Figure 3.
> Resizing CelebA to 32x32 allows us to train and evaluate our method within reasonable computational resources. We also note that deep generative learning under DP constraints is a very difficult task, which is only just starting to be explored by the community. It is significantly more challenging than training regular generative models. To the best of our knowledge, our paper and Datalens [3] are the only works that have attempted generation of RGB images under strict DP settings at all, while all earlier works have experimented exclusively with grayscale images. Regarding Datalens and our DP-Sinkhorn, only we are able to generate plausible CelebA images (see Figure 3).
>
>     Scaling differentially private generative modeling up to higher resolutions and more complex images is a very important research direction. For example, architectures or objectives tailored to the DP setting may be explored or novel DP mechanisms that are better suited for generative models could be investigated.
>
> 6. **Regarding Mode collapse**, we see in experiments that DP-Sinkhorn delivers the highest classification accuracy to downstream classifiers among the tested methods. Yet the visual quality of its generated images appears similar to GS-WGAN (Figure 2). Thus, we deduce that samples generated by DP-Sinkhorn are more diverse (less mode-collapsed), which aids in the generalization of the trained classifiers. Generally, it is plausible that our training objective based on a primal form of the Wasserstein distance leads to less mode dropping than adversarial approaches, which are known to be prone to mode collapse.
>
> 7. **Regarding performance on unbalanced datasets**, it has been shown [4,5,6] that performance of DP models on unbalanced datasets are disproportionately worse than their non-DP counterparts, suggesting fundamental incompatibility between differential privacy and fairness. It has also been shown that even WGAN-GP still exhibits non-convergence during training [7], which may be exacerbated on unbalanced datasets. We expect DP-Sinkhorn to fare better due to its stable training and diverse samples.
>
> We will improve the presentation in the final version of the paper to better discuss some of these aspects. Please let us know of any other questions or concerns. We would be very happy to discuss them. __We would like to kindly ask the reviewer to raise the review score, if they feel that our response adequately addresses their concerns.__ Thank you very much.
>
>
>
> [1] Bousquet, Olivier, et al. "From optimal transport to generative modeling: the VEGAN cookbook." arXiv preprint arXiv:1705.07642 (2017).
>
> [2] Xu, Lei, et al. "Modeling tabular data using conditional gan." arXiv preprint arXiv:1907.00503 (2019).
>
> [3] B. Wang, F. Wu, Y. Long, L. Rimanic, C. Zhang, and B. Li, “Datalens:  Scalable privacy pre-serving training via gradient compression and aggregation,”arXiv preprint arXiv:2103.11109,2021.
>
> [4] R. Cummings, V. Gupta, D. Kimpara, and J. Morgenstern, “On the compatibility of privacyand fairness,” inAdjunct Publication of the 27th Conference on User Modeling, Adaptationand Personalization, UMAP’19 Adjunct, (New York, NY, USA), p. 309–315, Association forComputing Machinery, 2019.
>
> [5] S. Kuppam, R. McKenna, D. Pujol, M. Hay, A. Machanavajjhala, and G. Miklau, “Fair decisionmaking using privacy-protected data,”CoRR, vol. abs/1905.12744, 2019.
>
> [6] Farrand, Tom, et al. "Neither private nor fair: Impact of data imbalance on utility and fairness in differential privacy." Proceedings of the 2020 Workshop on Privacy-Preserving Machine Learning in Practice. 2020.
>
> [7] Sanjabi M, Ba J, Razaviyayn M, Lee JD. On the convergence and robustness of training gans with regularized optimal transport. arXiv preprint arXiv:1802.08249. 2018 Feb 22.

---

### Author Response · Authors · 2021-09-02
**Post-Rebuttal Follow-Up**

We would like to follow up with all reviewers regarding the rebuttal process, since we did not hear back from any reviewer. We do hope, however, that we were able to address all questions and concerns in our detailed responses. We would incorporate all clarifications in the final version of the paper. If you feel that our responses are satisfactory, we would like to kindly ask you to consider raising your paper ratings accordingly. Otherwise, please let us know about the remaining concerns and where you think we could further improve. Such post-rebuttal feedback would be greatly appreciated. Thank you very much!

---

### Decision · Program_Chairs · 2021-09-27

**Decision:**

Accept (Poster)

**Comment:**

This paper proposes an DP algorithm for generative modeling, based on Sinkhorn divergence. This approach is non-adversarial, leading to a simpler and more stable optimization problem that is better suited for making differentially private. On the other hand, empirically, even non-private version of non-adversarial generative models lead to worse models and this limits the possible impact of this work.

Papers at the intersection of two active areas of research are usually hard to evaluate and this paper is no exception. The reviewers were divided on this work. The improvement over previous work is small in FID score, though FID is not a particularly good measure. The generated images are reasonably good.

I find the approach promising and the authors might want to evaluate if this approach leads to more "stable" optimization compare to GAN-based approaches; e.g. is this algorithm less sensitive to the choice of hyperparameters. The authors are also encouraged to evaluate this approach on medical datasets, such as those used in https://doi.org/10.1101/159756

On balance, this paper may lead to a broader range of ideas being explored in this space, and therefore I would recommend acceptance.